# ReST-RL: Reinforcing LLM Reasoning through Self-Training and Value-Guided Decoding

## Abstract

With respect to improving the reasoning accuracy of LLMs, the representative reinforcement learning (RL) method — Group Relative Policy Optimization (GRPO) — has achieved critical success, yet it still suffers from the issue of insignificant reward variance. This paper introduces **ReST-RL**, a unified LLM RL paradigm that combines an improved GRPO algorithm with a meticulously designed test-time decoding method to improve LLM's code reasoning ability. As the first stage of policy reinforcement, *ReST-GRPO* adopts an optimized ReST algorithm to increase the reward variance of GRPO sampling, thereby improving training effectiveness. Building on this foundation, we further introduce a test-time decoding optimization method, *VM-MCTS*, which employs an adapted Monte-Carlo Tree Search (MCTS) guided by a trained Value Model (VM) to provide precise process signals and verification scores, further enhancing LLM reasoning accuracy. We validate our RL paradigm on multiple coding benchmarks (e.g., APPS, BigCodeBench, and HumanEval), where it significantly outperforms other reinforcement training baselines (e.g., naive GRPO and ReST-DPO), as well as decoding and verification baselines (e.g., PRM-BoN and ORM-MCTS), indicating its power to strengthen LLM's reasoning capability. We further examine ReST-RL on out-of-domain math reasoning tasks, demonstrating that ReST-RL and the VM have strong transferability and generalizability across unseen reasoning domains and policy checkpoints, confirming that it extends beyond coding. Notably, our approach achieves strong performance with limited data, showcasing its effectiveness, efficiency, and generalizability.

## 1 Introduction

Recent advances in Large Language Models (LLMs) have made remarkable progress in reasoning tasks [20; 6; 39], yet substantial challenges remain for solving complex ones [7; 41]. Reinforcement Learning (RL) has emerged as a promising approach to further improve the reasoning ability of LLMs [18; 28; 22; 5; 21; 23; 24; 35], with growing attention on both online methods like Group Relative Policy Optimization (GRPO) [23] that data sampling and model updating take place simultaneously [22; 21; 23], and offline self-training paradigms like Reinforced Self-Training (ReST) [5; 35] that obtains training data through offline sampling and filtering mechanisms [5; 24; 35]. In addition, the reward models (RMs) are gradually attracting more attention due to their function in output verification, with outcome reward models (ORMs) that verify the final output of LLMs [27; 14] and process reward models (PRMs) that provide feedback for intermediate steps and indicate better verification accuracies than ORMs [14; 28; 39; 35], improving reasoning accuracy of LLMs. Moreover, integrating RMs with advanced decoding algorithms, such as Monte-Carlo Tree Search (MCTS), has shown further enhancements in LLM reasoning performance [35; 16].

However, there are still some shortcomings in these methods. On one hand, online RL algorithms represented by GRPO often lead to *unsatisfactory training results due to insignificant differences in reward signals* [33; 37]. Although some approaches attempt to mitigate this problem through specially designed step-wise rewards or simple dynamic sampling [33; 37], they lead to higher training computation costs and insufficient generalizability, also increasing the complexity of the RL algorithm. On the other hand, although PRMs can validate the output of LLMs more accurately than ORMs, they often require *high quality annotated data* [14; 28], which is costly and labor-intensive to obtain, poses significant challenges for scaling PRM training and limits their accuracy and reliability. To tackle the problem, some methods propose to estimate and collect process rewards through Monte-

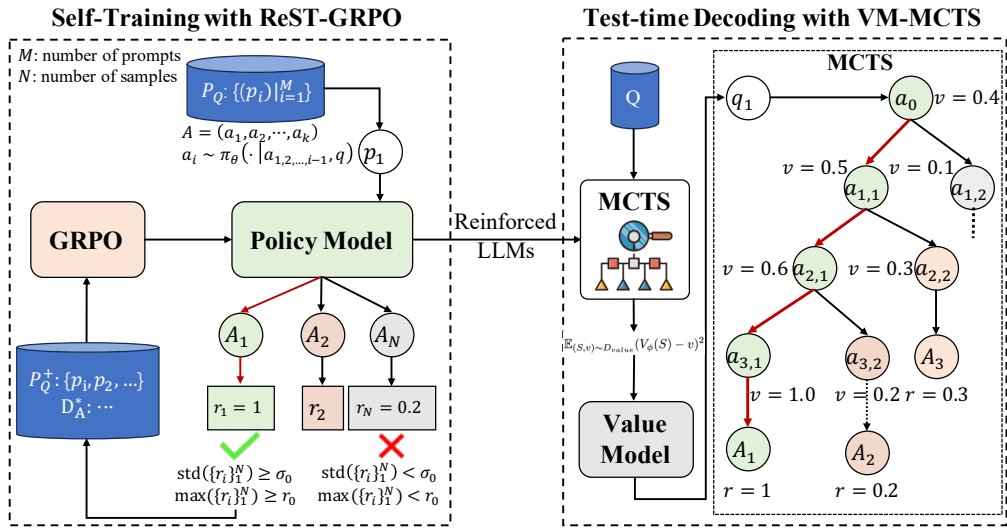

Figure 1: Framework of ReST-RL. First, ReST-GRPO adopts an optimized ReST algorithm to filter and assemble high-value training data. Subsequently, VM-MCTS optimizes LLM test-time decoding through process signals and verification scores provided by VM and MCTS.

Carlo simulations [28; 35]. Nevertheless, these methods fail to generalize to richer reasoning tasks such as coding, and their reliance on result matching mechanisms limits the scenarios in which they can be applied [35]. We provide a more comprehensive review of these LLM reinforcement methods (see Appendix A.2). Overall, although current LLM reinforcement methods have been attempted in terms of online training and output verification, they cannot simultaneously guarantee *low data collection costs, decent generalization, satisfactory reinforcement results, and training efficiency*.

In this paper, we introduce ReST-RL, a unified LLM RL framework that can significantly improve LLM's reasoning ability while balancing other factors. Our approach proposes to enhance LLMs through two stages, leveraging both an improved online RL algorithm and verified decoding, i.e., ReST-GRPO and VM-MCTS. As the first stage of LLM reinforcement, ReST-GRPO adopts an optimized ReST algorithm to filter and assemble high-value training data, increasing the reward variance of GRPO sampling, thus improving the effectiveness and efficiency of training. As the second stage of reinforcement, VM-MCTS optimizes test-time decoding through an elaborately designed value model (VM) and MCTS. Estimated value targets are collected using MCTS and a reward function without any extra annotation, enabling training of the VM at low cost. When decoding, the VM is deployed by an adapted MCTS algorithm to provide precise process signals as well as verification scores, assisting the LLM to achieve high reasoning accuracy. We validate the effectiveness of the proposed RL paradigm and its components through extensive experiments on well-known coding benchmarks of various levels (e.g., APPS [7], BigCodeBench [41], and HumanEval [3]), as well as some challenging mathematical reasoning benchmarks (e.g., MATH [8] and Omni-MATH [4]). Upon comparison, our approach significantly outperforms other reinforce training baselines such as naive GRPO and ReST-DPO, as well as decoding and verification baselines, such as PRM-BoN and ORM-MCTS. We show that ReST-RL has the power to strengthen the reasoning ability of LLM policies while balancing other aspects such as efficiency, cost, and generalizability.

To summarize, our contributions are:

- We introduce ReST-RL, an LLM RL paradigm that enhances LLM's reasoning ability by ReST-GRPO and VM-MCTS. The highlight of the method is that it combines the advantages of offline self-training, online learning algorithms, optimized decoding algorithms, and verification methods, achieving balanced efficiency, generalizability, cost, and effectiveness.

- We show that ReST-RL and its two key components outperform other reinforcement training baselines (e.g., naive GRPO and ReST-DPO, in Table 1) and decoding and verification baselines (e.g., PRM-BoN and ORM-MCTS, in Table 2).

- By comparing the test performance under the same training steps, ReST-GRPO has higher training efficiency compared to naive GRPO and DAPO (Figure 3 (a)). Given the same budget for decoding verification, VM-MCTS and its VM achieve better accuracy than previous Math-Shepherd style PRM or ORM trained on curated public data (Figure 3 (b)).

## 2 THE ReST-RL METHOD

In order to unify RL algorithms and decoding methods for LLMs and to address the issues of training reward variance and PRM accuracy, we propose a new approach, i.e., ReST-RL. It is composed of two main components, namely ReST-GRPO and VM-MCTS, as shown in Figure 1.

- *ReST-GRPO* is the first stage of learning, which adopts an optimized ReST algorithm to perform GRPO to strengthen the policy's capabilities on complex reasoning tasks.
- *VM-MCTS* is the second stage of reinforcement, which focuses on training and utilizing a value model (VM) that assists the policy in test time decoding.

### 2.1 TASK SETUP

For a specific reasoning task, there is generally an instruction or question, denoted as $q$, which describes the background information and problem details in text. There may exist a ground truth answer $g$. For coding tasks, there may also be some test cases $t_{1,2,\ldots,m}$. Using an LLM policy known as $\pi_\theta$, we can gradually generate solutions to the problem, following the predicted output token probabilities of the policy. This process can be regarded as a Markov Process, where the policy takes an action $a_i$ step by step following $\pi_\theta(a_i|a_{1,2,\ldots,i-1}, q)$, based on previously generated content $a_1, a_2, \ldots, a_{i-1}$ and the instruction $q$. Here, an action can represent a single token, a line of text, or even a reasoning step. Eventually, when the eos token or stop strings are generated, the process stops, forming a final solution $A = (a_1, a_2, \ldots, a_k)$.

In our approach, to align with the definitions in common RL, we define an action as a single line of text, which is separated by a line break. We also define an intermediate state as the combination of the instruction and a partial solution that contains a few lines of text, i.e. $S_i = (q, a_{1,2,\ldots,i})$. This means an intermediate state always terminates with a line break. In comparison, an end state stands for the combination of the instruction and a full solution that terminates with an eos token, i.e. $S_{end} = S_k = (q, a_{1,2,\ldots,k})$. In consistency with conventional settings for reasoning scenarios, the transition to the next state after an action was taken is considered to be deterministic. For our case, this means a new line of solution is appended to the current partial solution.

In terms of reward, we assign rewards for a reached state only when it is an end state. The function that determines the reward for an end state is denoted by $R = R(S_{end})$, which can be either rule-based or model-based (ORM, PRM). Moreover, the value function, or the expectation of reward under a certain policy, is defined by Eq. (1).

$$V^\pi(S_i) = \mathbb{E}_\pi[R(S_{end})|S_i]. \tag{1}$$

Under this setting, the state-action value function (Q function) can also be unified to the value function, as illustrated in Eq. (2).

$$Q^\pi(S_i, a_{i+1}) = \mathbb{E}_\pi[R(S_{end})|S_i, a_{i+1}] = \mathbb{E}_\pi[R(S_{end})|S_{i+1}] = V^\pi(S_{i+1}). \tag{2}$$

### 2.2 ReST-GRPO

Inspired by the discoveries of ReST [5] and GRPO [23] algorithms, we propose Reinforced Self-Training with Group Relative Policy Optimization (ReST-GRPO), a novel LLM RL algorithm that improves the efficiency and effectiveness of GRPO training through a few sampling and filtering mechanisms. As a crucial first step of our paradigm, ReST-GRPO leverages the policy itself to provide insight into train data selection and assembly, mitigating the reward failure problem of GRPO [33]. It strengthens the capability of the policy to generate reliable reasoning traces, getting it prepared for the second stage of reinforcement.

ReST-GRPO is designed to be a iterative process that reinforces the policy through multiple self-train iterations. In each iteration, the training process deploys the policy state at the end of last round. The

main algorithm of ReST-GRPO is composed of three steps within each reinforce cycle: (1) pre-train solution sampling, (2) train data filtering and assembly, (3) training with the GRPO objective. We present it in Algorithm 1.

---

**Algorithm 1:** ReST-GRPO algorithm for policy training.

---

**Input:** base LLM policy $\pi_\theta$, reward function (model) $R$, original question dataset $Q$, original prompt dataset $P_Q$, number of solution samples $N$, standard deviation threshold $\sigma_0$, reward threshold $r_0$, ratio of data to sample $\beta$, sampling exponent factor $\alpha$, number of iterations $T$.

1: **for** $i = 1$ to $T$ **do**
2:    $P_Q^+ \leftarrow \varnothing$ // initialize train set
3:    **for** prompt $p$ in $P_Q$ **do**
4:      Sample $N$ solutions $\{A_i\}_{i=1}^N \sim \pi_\theta(\cdot|p)$ // generate solutions samples
5:      Get reward $\{r_i = R(p, A_i)\}_{i=1}^N$ // obtain rewards for samples
6:      $\sigma \leftarrow std(\{r_i\}_{i=1}^N)$ // calculate standard deviation for reward
7:      **if** $\sigma \geq \sigma_0$ **then**
8:        $P_Q^+ \leftarrow p$
9:        **if** $max_i(r_i) \geq r_0$ **then**
10:         $A \leftarrow argmax_{A_i}(r_i)$ // get the solution with highest reward
11:         $D_A \leftarrow \{a_{1,2,\dots,j}\}_{j=1}^{|A|}$ // extract partial solutions
12:         $D_A^* \leftarrow do\_sample(D_A, \beta, \alpha)$ // sample partial solutions for train data assembly
13:         **for** partial solution $p^*$ in $D_A^*$ **do**
14:           $P_Q^+ \leftarrow p + p^*$
15:         **end for**
16:        **end if**
17:      **end if**
18:    **end for**
19:    $\pi_\theta \leftarrow GRPO(P_Q^+, \pi_\theta, R)$ // train the policy with the GRPO objective
20: **end for**
**Output:** $\pi_\theta$

---

### 2.2.1 PRE-TRAIN SOLUTION SAMPLING

Similar to the ReST algorithm that augments the initial training dataset with samples from the policy itself, ReST-GRPO also utilizes the sample solutions generated from the policy for further train data selection and assembly. Within each iteration, we first collect $N$ solutions for every instruction prompt in the original dataset using the current policy. By configuring the LLM's sampling temperature, we can control the randomness and diversity of these solutions. Then, we utilize a fixed reward function to obtain rewards for all solutions, which are used for the subsequent filtering process.

### 2.2.2 DATA FILTERING BY REWARD

An LLM policy's output solutions and their corresponding rewards contain a wealth of information indicating its strength and weakness toward the target domain, which can be leveraged to filter out ineffective train data. For GRPO, policy update relies on group relative advantage. This means effective training signals depend on discrepancies between rewards, offering three key enlightenments:

- If a policy's output solutions obtain similar rewards on a question, there's nothing much for it to learn from this question.

- For questions or tasks where the current policy faces an enormous action space but only few output traces lead to a substantial reward, normal sampling from the initial state may be ineffective for training.

- For a question that the current policy does not perform well, high-reward solution traces are crucial for training. Since high-reward solutions often share some common patterns, sampling from a

partial solution state of a high-reward solution may be helpful for obtaining more high-reward traces, thus beneficial for training.

We design our algorithm's filtering and train data assembly process based on these enlightenments. Firstly, we use standard deviation to evaluate the variety of rewards. For question prompts that the policy's solutions achieve a reward standard deviation less than a given threshold $\sigma_0$, we filter them out from the train dataset, because they will possibly result in very little improvement of the policy. For other prompts, we add them to the train dataset. Secondly, we focus on high-reward solution traces that may be beneficial for training. Specifically, we filter out question prompts that the policy only achieve rewards lower than a certain threshold $r_0$. For other prompts, we extract their corresponding solution trace with the highest reward. Finally, its partial solution states are utilized to assemble new train data, as illustrated in Section 2.2.3. In addition, we present in Figure 2 the comparison of training reward variances between ReST-GRPO and naive GRPO, which validates the effect of ReST-GRPO in improving the variance of rewards.

### 2.2.3 TRAIN DATA ASSEMBLY

Considering that a partial state belonging to a high-reward solution may possibly be a good state for the policy to begin with, we extract a subset of all these partial solution states for new train data assembly. For simplicity, we only select from partial states of the solution trace $A$ with the highest reward for each prompt $p$. Denoted by $a_{1,2,\ldots,j}$, a partial state contains the first $j$ actions (i.e. lines in our setting) of $A$ taken by the policy. In the assembly process, we first sample a subset of these partial states $D_A^*$ based on a discrete finite exponential distribution, using a fixed positive exponent factor $\alpha < 1.0$. Similar to a geometric distribution, except that the random variable only takes a finite number of values, we sample each time following the probability mass function defined by Eq. (3). This p.m.f is designed to control the diversity of reward, maintaining some probability for the policy to start from partial states that are close to a high-reward final solution during training, while assigning more probability weight to initial states. Using this p.m.f, we sample $\beta|A|$ partial states and add them to $D_A^*$, where $\beta$ is a preconfigured ratio parameter. Then, every partial state within $D_A^*$ is appended to the initial prompt $p$, forming new train data. Note that our method is similar to common GRPO when $\alpha$ is close to 0, since it only uses the initial prompt state for training.

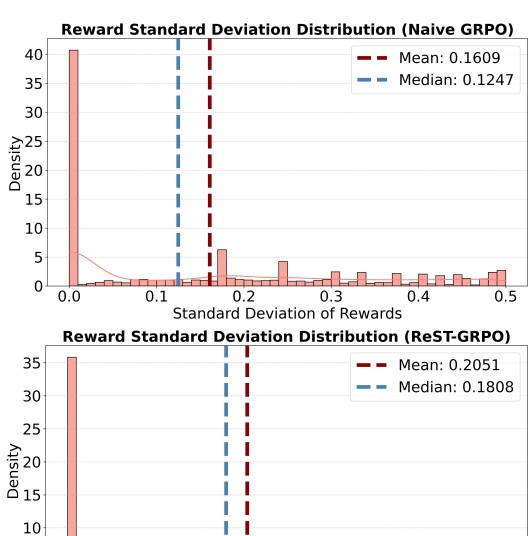

Figure 2: Distribution of standard deviation of group rewards during policy training. We can observe a clear distributional shift between two algorithms. Training configurations are illustrated in Section 3.2, and the base LLM policy here is OpenCodeInterpreter-DS-6.7B.

$$p(a_{1,2,\ldots,j}) = \frac{1-\alpha}{1-\alpha^{|A|}}\alpha^{j-1}, j = 1, 2, \ldots, |A| \tag{3}$$

### 2.2.4 GRPO TRAINING

After training data for each iteration are obtained, online reinforce training is performed. Policy update is based on the objective displayed in Eq. (4), which is basically the common GRPO objective, except that the policy's input prompt can be a combination of question and sampled partial solution

from the train set $P_Q^+$.

$$\mathcal{J}(\theta) = \mathbb{E}[p \sim P_Q^+, \{o_i\}_{i=1}^G \sim \pi_{\theta_{old}}(O|p)]$$

$$\frac{1}{G}\sum_{i=1}^G \frac{1}{|o_i|}\sum_{t=1}^{|o_i|}\{min[\frac{\pi_\theta(o_{i,t}|p, o_{i,<t})}{\pi_{\theta_{old}}(o_{i,t}|p, o_{i,<t})}\hat{A}_{i,t}, clip(\frac{\pi_\theta(o_{i,t}|p, o_{i,<t})}{\pi_{\theta_{old}}(o_{i,t}|p, o_{i,<t})}, 1-\epsilon, 1+\epsilon)\hat{A}_{i,t}] \quad (4)$$

$$- \beta'\mathbb{D}_{KL}[\pi_\theta||\pi_{ref}]\}$$

## 2.3 VM-MCTS

In this section, we present Value Model based Monte-Carlo Tree Search (VM-MCTS), an LLM test-time decoding reinforcement method that balances exploration and exploitation during decoding process through an adapted MCTS algorithm, while also boosting the policy output's accuracy and reliability via an elaborately designed VM. By performing MCTS with a reward function, accurate process value targets are collected, enabling the training of a VM. As an essential component for test-time MCTS-based decoding, the VM plays a similar role as a PRM. It not only provides verification signals, but also guides the LLM policy to search for high potential reasoning traces.

VM-MCTS is a continuation and expansion of the previous Math-Shepherd (M-S) method [28]. We generalize the M-S style PRM to our definition of VM. Although the value target introduces some extra noise, it actually leads to better performance when decoding, with the assistance of MCTS.

### 2.3.1 THE VALUE TARGET

In the M-S method, the quality of an intermediate step is defined as its potential to deduce the correct final answer, which is an expectation-based evaluation target, resembling the value function. For VM-MCTS, we summarize this to the value target defined in Eq. (1). Rather than evaluating a single step or action, the value target emphasizes the evaluation of the whole partial state including the last action. It naturally reflects the potential for a policy to reach high-reward end states from the current partial state, which can be leveraged to assist policy decoding. In fact, the M-S method can be regarded as a special case where the reward function $R$ is defined by Eq. (5).

$$R(S_{end}) = \mathbb{I}(a_A = a^*),$$
$$s.t. \quad S_{end} = (q, A) \quad (5)$$

### 2.3.2 VALUE MODEL TRAINING

We adopt MCTS when collecting training data for the VM, as it balances exploration of different reasoning paths and exploitation of promising intermediate states. In our MCTS algorithm, the root node of the search tree is set to the initial state $S_{init} = S_0 = (q, \varnothing)$, while other nodes represent intermediate states or end states $S_i = (q, a_{1,2,\ldots,i})$. As illustrated in Section 2.1, an action represents a line of text. Through multiple rounds of MC simulation, value targets of partial states and end states are gradually updated and precisely estimated, distinguished from the all-at-once estimation used by M-S. We present the pseudocode for this process in Algorithm 2. Once sufficient value data are acquired, a VM $V_\phi$ is trained on these data to predict the value target of different states. The loss function used by VM-MCTS in the training process is demonstrated in Eq. (6).

$$\mathcal{L}_\phi = \mathbb{E}_{(S,v)\sim D_{value}}(V_\phi(S) - v)^2 \quad (6)$$

### 2.3.3 ASSISTED DECODING

Previous approaches like PPO-MCTS [16] have demonstrated the effectiveness of conventional VMs and MCTS in token-level assisted decoding process. For our case, a VM fully trained with our method can accurately predict the expected reward of partial states based on current policy, as well as the determined reward of end states, indicating its potential for assisted decoding. For VM-MCTS, we use an adapted version of MCTS to assist policy decoding at test time. Deploying our VM, we are also able to perform verification in the decoding process, similar to common PRMs. As shown in Algorithm 3, the adapted search algorithm deploys the VM to provide value signals for MC rollouts, while also verifying and selecting solutions by Best-of-N as previous verification methods do [14]. The algorithm uses value estimations to decide which paths to decode and explore, which improves

Table 1: Main results for policy training. LLM policies are evaluated after each training iteration ends. We use the same sampling temperature of $0.7$ for every LLM when testing. In the table, "iter" refers to the number of training iterations performed on the LLM on $Q_{train}$ (0th iter. means not trained).

| Model | Training Method | HumanEval | HumanEval+ | MBPP | MBPP+ | APPS-500 | BCB | Avg. |
|---|---|---|---|---|---|---|---|---|
| | Base (0th iter.) | 0.829 | 0.78 | 0.717 | 0.622 | 0.118 | 0.418 | 0.503 |
| | Below are results for sequential training iterations | | | | | | | |
| Qwen3-8B | ReST-DPO (1st iter.) | 0.854 | 0.799 | 0.73 | 0.627 | 0.152 | 0.434 | 0.523 |
| | GRPO (1st iter.) | 0.799 | 0.75 | 0.754 | 0.651 | 0.346 | 0.439 | 0.566 |
| | **ReST-GRPO (1st iter.)** | 0.872 | 0.817 | 0.78 | 0.672 | 0.377 | 0.469 | 0.604 |
| | ReST-DPO (2nd iter.) | 0.841 | 0.811 | 0.741 | 0.653 | 0.153 | 0.429 | 0.526 |
| | GRPO (2nd iter.) | 0.829 | 0.787 | 0.757 | 0.667 | 0.403 | 0.436 | 0.590 |
| | **ReST-GRPO (2nd iter.)** | 0.86 | 0.805 | 0.802 | 0.69 | 0.565 | 0.476 | **0.655** |
| | Base (0th iter.) | 0.872 | 0.799 | 0.817 | 0.683 | 0.287 | 0.381 | 0.563 |
| | Below are results for sequential training iterations | | | | | | | |
| Qwen2.5-Coder-7B-Instruct | ReST-DPO (1st iter.) | 0.884 | 0.823 | 0.820 | 0.688 | 0.321 | 0.386 | 0.579 |
| | GRPO (1st iter.) | 0.896 | 0.835 | 0.812 | 0.693 | 0.350 | 0.405 | 0.593 |
| | **ReST-GRPO (1st iter.)** | 0.872 | 0.817 | 0.823 | 0.701 | 0.401 | 0.441 | 0.612 |
| | ReST-DPO (2nd iter.) | 0.872 | 0.823 | 0.796 | 0.669 | 0.330 | 0.400 | 0.578 |
| | GRPO (2nd iter.) | 0.872 | 0.799 | 0.815 | 0.685 | 0.356 | 0.403 | 0.586 |
| | **ReST-GRPO (2nd iter.)** | 0.890 | 0.835 | 0.849 | 0.712 | 0.415 | 0.462 | **0.630** |
| | Base (0th iter.) | 0.744 | 0.677 | 0.741 | 0.646 | 0.230 | 0.338 | 0.493 |
| | Below are results for sequential training iterations | | | | | | | |
| DS-Coder-6.7b-Instruct | ReST-DPO (1st iter.) | 0.756 | 0.683 | 0.725 | 0.635 | 0.251 | 0.329 | 0.495 |
| | GRPO (1st iter.) | 0.726 | 0.646 | 0.735 | 0.632 | 0.262 | 0.342 | 0.493 |
| | **ReST-GRPO (1st iter.)** | 0.756 | 0.683 | 0.743 | 0.635 | 0.282 | 0.355 | 0.511 |
| | ReST-DPO (2nd iter.) | 0.756 | 0.689 | 0.720 | 0.632 | 0.242 | 0.329 | 0.492 |
| | GRPO (2nd iter.) | 0.707 | 0.634 | 0.730 | 0.624 | 0.283 | 0.357 | 0.497 |
| | **ReST-GRPO (2nd iter.)** | 0.793 | 0.707 | 0.749 | 0.646 | 0.300 | 0.368 | **0.529** |
| | Base (0th iter.) | 0.756 | 0.707 | 0.722 | 0.630 | 0.204 | 0.331 | 0.486 |
| | Below are results for sequential training iterations | | | | | | | |
| OpenCI-DS-6.7B | ReST-DPO (1st iter.) | 0.756 | 0.677 | 0.714 | 0.622 | 0.212 | 0.351 | 0.487 |
| | GRPO (1st iter.) | 0.726 | 0.677 | 0.733 | 0.632 | 0.277 | 0.363 | 0.506 |
| | **ReST-GRPO (1st iter.)** | 0.726 | 0.677 | 0.725 | 0.635 | 0.281 | 0.374 | 0.509 |
| | ReST-DPO (2nd iter.) | 0.744 | 0.671 | 0.725 | 0.630 | 0.232 | 0.336 | 0.488 |
| | GRPO (2nd iter.) | 0.750 | 0.689 | 0.743 | 0.648 | 0.279 | 0.352 | 0.512 |
| | **ReST-GRPO (2nd iter.)** | 0.774 | 0.713 | 0.725 | 0.630 | 0.325 | 0.377 | **0.531** |

Table 2: Average results of ReST-RL and different verification methods on all benchmarks. For all LLM policies, the sampling temperature is set to $0.7$. All verification is based on $100$ samples.

| Method | Qwen3-8B | Qwen2.5-Coder-7B-Instruct | DS-Coder-6.7b-Instruct | OpenCI-DS-6.7B |
|---|---|---|---|---|
| Base | 0.503 | 0.563 | 0.493 | 0.486 |
| ORM | 0.531 | 0.592 | 0.542 | 0.537 |
| PRM | 0.516 | 0.591 | 0.539 | 0.532 |
| ORM-MCTS | 0.538 | 0.588 | 0.547 | 0.535 |
| VM-MCTS | 0.615 | 0.652 | 0.576 | 0.569 |
| ReST-RL | **0.689** | **0.673** | **0.584** | **0.583** |

search efficiency and accuracy. We also prove in Appendix A.8 that our value-based MC rollout estimates partial state values more precisely than the usual complete trace rollout method, resulting in better search performance.

## 3 EXPERIMENTS

As an important branch of LLM reasoning tasks, code writing has received much attention in recent years [38; 13; 11]. In this research, we also focus on coding problems, considering that they have a more comprehensive test of LLM's reasoning, planning, and invocation abilities. In this section, we present various experiments to validate ReST-RL. As two main components of ReST-RL, ReST-GRPO and VM-MCTS are separately verified through comparison to baseline training or decoding methods on multiple well-known benchmarks such as HumanEval [3], MBPP [1] and APPS [7]. As a

whole, we demonstrate that the combination of these two components leads to optimal test results, justifying our overall approach.

## 3.1 EXPERIMENT SETTINGS

We present the main experiment settings in this section, including selected LLMs, datasets used for training and the base reward function. Additionally, more experimental details for benchmark evaluation are displayed in Appendix A.3.

**Base LLM Policies.** In our experiment, we primarily test the proposed method with four recent code LLMs, namely Qwen2.5-Coder-7B-Instruct [9], CodeQwen1.5-7B-Chat [25], Deepseek-Coder-6.7B-Instruct [6] and OpenCodeInterpreter-DS-6.7B [40]. Although these LLMs already have decent capabilities in coding tasks, we show that ReST-RL can still significantly improve their programming skills. To further testify our method for general LLMs, we also include Qwen3-8B [26], Llama-3-8B and Llama-3.1-8B-Instruct [19] in our experiments, indicating the generalizability of our approach.

**Training Datasets.** In terms of source datasets for training, we select three well-known open-source coding datasets: BigCodeBench (BCB) [41], DS-1000 [10] and APPS [7]. After filtering out data in the source train sets without test cases, we obtain a final dataset $Q_{train}$ of 6945 coding prompts. For experiments mentioned in Section 3.2 and Section 3.3, training of policies and value models with ReST-RL and other baselines are all based on this dataset.

**Base Reward Function.** According to previous research, the use of model-based learned reward functions in reinforcement learning scenarios may increase the risk of reward hacking [30], causing undesirable effects on our experimental results. Therefore, we primarily adopt a rule-based reward function that uses test cases to calculate reward, as shown in Eq. (7). It evaluates generated codes with the ratio of passed test cases to total test cases, which naturally reflects the functional correctness of these codes. Somewhat differently, two additional rewards have been added to improve stability and effectiveness for ReST-GRPO and naive GRPO training, which is illustrated in Appendix A.5.

$$R_{base}(S_{end}) = R_{base}(q, A) = \frac{1}{m} \sum_{i=1}^{m} \mathbb{I}(\text{eval}(A, t_i) = y_i), \tag{7}$$

where $t_1, t_2, \ldots, t_m$ are test cases, and $y_1, y_2, \ldots, y_m$ are desired test outputs.

## 3.2 EVALUATION OF LLM POLICY TRAINING METHODS

In order to assess the effectiveness of ReST-GRPO, we conduct experiments on LLM policy training, comparing ReST-GRPO mainly to two other baselines: ReST-DPO (basically ReST$^{EM}$ [24] with DPO based optimization) and naive GRPO. The detailed configurations for these approaches are shown in Appendix A.5. Note that for all policies and methods, we adopt a sampling temperature of 0.7, and the learning rate is set $1e - 7$.

**Performance comparison for iterative policy training.** We compare ReST-GRPO with other baselines by performing 2 sequential training iterations on $Q_{train}$. Performance of four LLM policies after each training iteration on all benchmarks is shown in Table 1, and additional results for Llama-3.1-8B-Instruct and CodeQwen1.5-7B-Chat are shown in Appendix A.7. We discover that for all LLMs, ReST-GRPO significantly outperforms ReST-DPO and naive GRPO regarding average score for each iteration. After two iterations of training, ReST-GRPO achieves optimal improvements in average score of 15.2%, 6.7%, 3.6% and 4.5% respectively for the selected LLMs. Moreover, We also find that ReST-GRPO is the only approach which does not degrade on any of the benchmarks after two rounds of training.

**Assessment of training efficiency for ReST-GRPO, DAPO and naive GRPO.** By evaluating policy performance after same number of training steps, we can get a glimpse of the difference in training efficiency between ReST-GRPO, DAPO [33] and naive GRPO. In Figure 3 (a), we display average accuracy results of Llama-3-8B on all benchmarks within $10k$ training steps. We select Llama-3-8B as base policy for mainly two reasons: (1) As an LLM not specifically finetuned on coding tasks, there is more space for improvement in its coding capabilities. Differences in training effectiveness between methods may be more pronounced. (2) We would like to examine the effectiveness of our method for general base LLMs. In terms of results, we find that although in the first $2k$ training

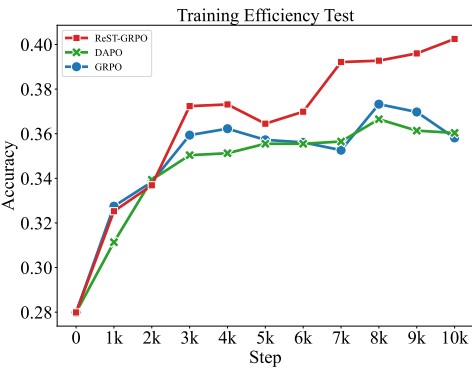 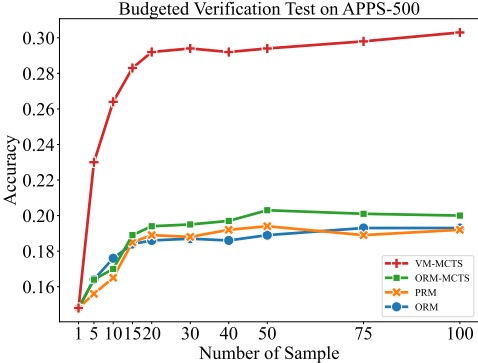

(a) Training efficiency assessment for ReST-GRPO, DAPO, and naive GRPO based on average score on all benchmarks. We adopt Llama-3-8B as the base policy, which is then trained for $10k$ steps using two methods. Policy performance is evaluated every $1k$ steps.

(b) Budgeted verification results based on CodeQwen. We evaluate the Test Case Average score of verified outputs on APPS-500, using different verification methods. Sampling temperature is set to $0.7$ for all methods.

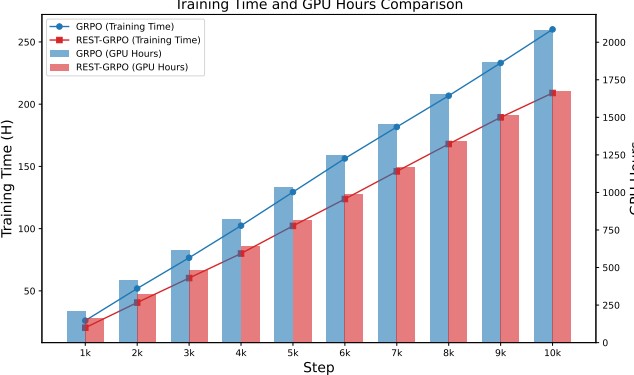

(c) Training time and corresponding GPU-hour accounting for Llama-3-8B, comparing ReST-GRPO with naive GRPO. The training time is recorded every 1000 steps.

Figure 3: Test of training efficiency and budgeted verification.

steps the performance of the three algorithms are comparable, ReST-GRPO's lead shows a gradual increase after $2k$ steps. At $10k$ steps, ReST-GRPO attains a substantial improvement of $12.3\%$, greatly outperforming $7.8\%$ achieved by naive GRPO, and $8.1\%$ achieved by DAPO. Moreover, Figure 3 (c) displays the training time and GPU-hours required for every $1k$ training samples for ReST-GRPO and naive GRPO. Due to ReST-GRPO's pre-train filtering and partial sampling design, it attains an approximately $20\%$ reduction on training time while achieving better training results, as shown Figure 3 (a). These results indicate both efficiency and sustainability of our method, implying its long term effectiveness on training.

## 3.3 ASSESSMENT OF DECODING AND VERIFICATION METHODS

In this part, we present experiment results that verify the superiority of VM-MCTS over some widely used decoding and verification methods, which are primarily based on ORMs and PRMs. In our experiments, we compare VM-MCTS with three other baselines: ORM+BoN (denoted by ORM), PRM+BoN (denoted by PRM) and ORM-MCTS. Since previous work [35; 31] has shown that MCTS is more effective compared to BFS and DFS based tree search algorithms, we do not include them as baselines. The main configurations are described below. For all methods, the decoding temperature is set to $0.7$. See detailed description of decoding and verification methods in Appendix A.6.

**Comparison of method accuracy on all benchmarks.** We test VM-MCTS and other baselines on various base LLM policies. In Table 2, we present average results on all benchmarks, with the number of samples for verification limited to $100$. Results show that VM-MCTS significantly surpasses

Table 3: Out of distribution generalization results of the VM and the ReST-RL method on code and math reasoning benchmarks, based on Qwen3-8B. Aside from the normal setting, we also test the initial VM (0th iter.) with the policy updated after two rounds of ReST-GRPO training, verifying the transferability of the VM when facing distribution shifts.

| Model | Method | APPS-500 | BCB | MATH | Omni-MATH |
|-------|--------|----------|-----|------|-----------|
| | Base (0th iter.) | 0.118 | 0.418 | 0.780 | 0.234 |
| | Base (0th iter.) + VM (0th iter.) | 0.415 | 0.471 | 0.828 | 0.238 |
| Qwen3-8B | ReST-GRPO (2nd iter.) + VM (0th iter.) | 0.630 | 0.496 | 0.862 | 0.246 |
| | ReST-GRPO (2nd iter.) + VM (2nd iter.) | **0.642** | **0.506** | **0.872** | **0.256** |

other baselines for every base LLM, achieving average improvement of 11.2, 8.9%, 8.3%, and 8.3% respectively for Qwen3-8B, Qwen2.5-Coder, DS-Coder and OpenCI. The lead of VM-MCTS over ORM-MCTS validates the effectiveness of our VM for the decoding process.

**Verification results based on controlled budget.** By comparing verification methods under various controlled search budgets, We can comprehensively examine their performance and trends in effectiveness. In Figure 3 (b), we show verification results on APPS-500 under different number of samples, based on CodeQwen. We find that our method substantially outperforms other baselines at all budgets and generally has a higher growth rate with increasing number of samples, again demonstrating the superiority of our method. To further reveal the computational overhead of VM-MCTS under various sampling sizes, we display the decoding token usage of VM-MCTS compared to conventional Best-of-N in Table 10, where $N$ is the total number of sampled traces. From the results we can conclude that VM-MCTS significantly outperforms Best-of-N across all sampling sizes while requiring fewer token computation, owing to its ability to perform rollouts from partial samples.

### 3.4 VALIDATION OF THE OVERALL APPROACH

Based on the final LLM policies trained with ReST-GRPO, we further test their performance on benchmarks after strengthened by VM-MCTS. As demonstrated in Table 2, the overall method achieves best average results for all base LLMs, surpassing separately used ReST-GRPO or VM-MCTS. The ablation results proves the necessity and effectiveness of the overall ReST-RL methodology.

**Generalization towards out-of-domain tasks.** In this part, we further investigate ReST-RL's performance and reliability on out-of-domain problems. We are also interested in exploring the robustness and transferability of the VM to unseen or biased states after significant policy updates, which introduces output distribution shifts. This is crucial for assessing the applicability and robustness of the proposed method. Therefore, we conduct out-of-domain testing of ReST-RL on mathematical reasoning tasks, represented by two well-known math reasoning benchmarks: MATH [8] and Omni-MATH [4]. In Table 3, we show the main results for Qwen3-8B, where ReST-RL achieves a noticeable improvement of 9.2% and 2.2% respectively for MATH and Omni-MATH. We can also observe that even after undergoing two full rounds of policy reinforcement learning, the initial VM (0th iter.) retains strong applicability. Although its score falls short of the similarly updated VM (2nd iter.), the performance loss is relatively minor (approximately 1%). These results demonstrate that ReST-RL and the VM have strong transferability and generalizability across reasoning domains and policy checkpoints. Note that this performance enhancement on MATH/Omni-MATH is achieved without any math-specific tuning, confirming that ReST-RL extends beyond coding.

## 4 CONCLUSION

In this paper, we introduce **ReST-RL**, a unified LLM RL paradigm that is capable of significantly improving the reasoning ability of LLMs by combining an improved GRPO algorithm with a meticulously designed test-time decoding method assisted by a VM. Through extensive experiments on various levels of coding and math reasoning tasks, we showcase the superiority of ReST-RL over other training and decoding baselines, indicating its potential to assist LLMs achieve breakthroughs in reasoning capabilities. Regarding future work, we would like to further investigate the generalizability of ReST-RL to a broader scope of tasks aside from math and code reasoning.

REPRODUCIBILITY STATEMENT

To promote reproducibility, we provide detailed pseudocode for the main algorithms in Appendix A.4. We also elaborate on the experimental setup and the compared methods in Section 3.1, Appendix A.5 and A.6. Additionally, we provide the codes related to this paper in the supplementary materials, which includes the main functions described in the paper.

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

# A  APPENDIX

## A.1  THE USE OF LARGE LANGUAGE MODELS (LLMS)

In this section, we declare the use of LLM. During the writing of this paper, we employed LLM to assist in creating illustrative figures for certain experimental results, primarily by generating the necessary code for plotting. In other aspects, such as research ideation and textual composition of the paper, we did not utilize LLM.

## A.2  RELATED WORK

**LLMs for reasoning and code synthesis.**  Recent advancements have facilitated the growing application of LLMs to reasoning tasks like coding [13; 6; 25; 40] and math [20; 23]. To assist LLMs on code synthesis, previous approaches use test cases to provide reward signals [38]. However, though demonstrating notable performance on elementary programming tasks, LLMs continue to struggle with complex ones, such as those found in competitive programming platforms like Codeforces [7].

**Improving reasoning abilities of LLMs.** Previous methods that attempt to enhance the reasoning abilities of LLM mainly divide into two paths: LLM training and optimized decoding. For the first path, aside from basic pretraining methods [20; 2], some suggest adopting supervised fine-tuning (SFT) with filtered web-source data [3] or self-generated data [24; 34]. To address the problem of insufficient data volume and expensive labeling, LLM self-training methods such as ReST have been proposed [5; 24; 35]. These methods use the LLM itself for data sampling, and after filtering these data are used for SFT. Alternatively, some approaches utilize reinforcement learning (RL) to achieve LLM enhancement through reward-based optimization algorithms [12; 11], represented by GRPO [23]. However, more recently, some researches have shown that GRPO may have problems in the effectiveness of its reward [37; 33]. As for the second path, prompting methods like Chain-of-Thought (CoT) are proved to be convenient and useful for decoding [29]. Tree search algorithms like Tree-of-Thought (ToT) [32] and Monte-Carlo Tree Search (MCTS) [35; 31; 36] are also proven to be effective when incorporated into the decoding process.

**Output verification with reward models.** Leveraging reward models, outputs of LLMs can be further verified, enabling selection of those that are more desirable and reliable. Existing reward models are divided into two main categories: (1) Outcome Reward Model (ORM) which provides an evaluation score for the entire output [39]. (2) Process Reward Model (PRM) [14; 27] which evaluates intermediate steps or actions. Despite outperforming ORMs in many situations [14; 27], PRMs require more high-quality process training data. Since PRM data annotation is expensive, some approaches try to estimate rewards through Monte-Carlo simulation [28; 35], which makes it possible to scale up PRMs, but also faces higher noise within reward data [39]. Moreover, although PRMs obtained by such methods can provide process signals, the essence of these signals is more closely related to expected values than step-level rewards [39]. Therefore, a more appropriate designation for these RMs is Value Model (VM) rather than PRM, and we also use this designation in this paper.

## A.3  BENCHMARK EVALUATION DETAILS

To comprehensively evaluate the effectiveness of different methods, we conduct test on six renowned benchmarks: HumanEval [3], HumanEval+ [17], MBPP [1], MBPP+ [17], BigCodeBench (BCB) [41] and APPS [7]. Among them, since the APPS test set is too large, we construct a new test set known as APPS-500 by randomly selecting 500 data from it, which is used as the final test dataset. As for evaluation metrics, we use pass@1 for all benchmarks except APPS-500. For APPS-500, we use the original Test Case Average metric proposed by the authors of APPS. In addition, all our evaluations use zero-shot performance.

## A.4  MAIN ALGORITHMS

We present the training data collection process of VM in Algorithm 2 and the assisted decoding algorithm of VM-MCTS in Algorithm 3.

---

**Algorithm 2:** The value model's train data collection process.

---

**Input:** LLM policy $\pi_\theta$, reward function (model) $R$, question (instruction) dataset $Q$, prompt dataset $P_Q$, max search iterations $T$, number of samples for each expansion $n$, exploration constant $c$, small constant that avoids zero-division $\epsilon$, initial value $v_0$,

1: $D_{value} \leftarrow \varnothing$ // initialize value train set
2: **for** prompt $p$ in $P_Q$ **do**
3:     Initialize the search tree $T_p$
4:     **for** $t = 1, 2, \ldots, T$ **do**
5:         $S \leftarrow S_{init}$
6:         ————————*Node Selection*————————
7:         **while** $S$ is expanded **do**
8:             $S \leftarrow argmax_{S' \in \text{children}(S)}(v_{S'} + c\sqrt{\frac{\ln N_S + 1}{N_{S'} + \epsilon}})$ // select child node based on UCT
9:         **end while**
10:         $v, m \leftarrow 0$ // for value calculation in backpropagation process
11:         **if** $S = (p, a_{1,2,\ldots,j})$ is not an end state **then**
12:             ————————*Node Expansion*————————
13:             Sample $n$ traces $\{o_i = (a_{j+1,j+2,\ldots}^{(i)})\}_{i=1}^n \sim \pi_\theta(O|S)$ starting from $S$
14:             Build child nodes by taking actions $\{a_{j+1}^{(i)}\}_{i=1}^n$ // use the first action of sample traces to expand
15:             **for** node $S'$ in children$(S)$ **do**
16:                 $v_{S'} \leftarrow v_0$
17:                 $N_{S'}, U_{S'} \leftarrow 0$
18:             **end for**
19:             ————————*MC Rollout*————————
20:             Get rewards with reward function $\{r_i = R(p, a_{1,2,\ldots,j}, o_i)\}_{i=1}^n$
21:             $D_{value} \leftarrow \{((p, a_{1,2,\ldots,j}, o_i), r_i)\}_{i=1}^n$ // add value targets for end states to train set
22:             $v, m \leftarrow sum(\{r_i\}_{i=1}^n), n$
23:         **else**
24:             $r \leftarrow R(S)$ // get the reward for current end state
25:             $v, m \leftarrow r, 1$
26:         **end if**
27:         ————————*Value Backpropagation*————————
28:         **for** node $S'$ from $S$ to root **do**
29:             $N_{S'} \leftarrow N_{S'} + m$ // update visit count
30:             $U_{S'} \leftarrow U_{S'} + v$ // update sum of reward
31:             $v_{S'} \leftarrow \frac{U_{S'}}{N_{S'}}$ // update value estimation
32:         **end for**
33:     **end for**
34:     **for** node $S$ in $T_p$ **do**
35:         **if** $S$ is expanded **then**
36:             $D_{value} \leftarrow (S, v_S)$ // add value targets for partial states to train set
37:         **end if**
38:     **end for**
39: **end for**
40: **Return** $D_{value}$
**Output:** $D_{value}$

---

## A.5   Details for Compared Policy Training Methods

- **ReST-GRPO:** In terms of reward function used during training, aside from the base reward evaluating functional correctness with test case pass rate, two additional rewards have been involved to improve training results, as demonstrated in Eq. (8). The first reward examines whether the output code contains certain essential substrings (such as identifiers for code segments), while the

---

**Algorithm 3:** The assisted decoding algorithm of VM-MCTS.

---

**Input:** LLM policy $\pi_\theta$, Value model $V_\phi$, original question (instruction) $q$, question prompt $p_q$, max search iterations $T$, number of samples for each expansion $n$, exploration constant $c$, small constant that avoids zero-division $\epsilon$,

1: Initialize the search tree $T_{p_q}$
2: $D_{sol} \leftarrow \varnothing$ // initialize solution set
3: **for** $t = 1, 2, \ldots, T$ **do**
4:   $S \leftarrow S_{init}$
5:   ————————*Node Selection*————————
6:   **while** $S$ is expanded **do**
7:     $S \leftarrow argmax_{S' \in \text{children}(S)}(v_{S'} + c\sqrt{\frac{\ln N_S + 1}{N_{S'} + \epsilon}})$ // select child node based on UCT
8:   **end while**
9:   $v, m \leftarrow 0$ // for value calculation in backpropagation process
10:   **if** $S = (p, a_{1,2,\ldots,j})$ is not an end state **then**
11:     ————————*Node Expansion*————————
12:     Sample $n$ traces $\{o_i = (a_{j+1,j+2,\ldots}^{(i)})\}_{i=1}^n \sim \pi_\theta(O|S)$ starting from $S$
13:     $D_{sol} \leftarrow \{(a_{1,2,\ldots,j}, o_i)\}_{i=1}^n$ // add sample solutions to solution set
14:     Build child nodes by taking actions $\{a_{j+1}^{(i)}\}_{i=1}^n$ // use the first action of sample traces to expand
15:     ————————*Value Based MC Rollout*————————
16:     **for** node $S'$ in children$(S)$ **do**
17:       $v_{S'} \leftarrow V_\phi(S')$ // get value estimation
18:       Set $N_{S'}$ to the number of times $S'$ was visited during expansion phase
19:       $U_{S'} \leftarrow N_{S'}v_{S'}$
20:       $v \leftarrow v + U_{S'}$
21:     **end for**
22:     $m \leftarrow n$
23:   **else**
24:     $v, m \leftarrow v_S, 1$
25:   **end if**
26:   ————————*Value Backpropagation*————————
27:   **for** node $S'$ from $S$ to root **do**
28:     $N_{S'} \leftarrow N_{S'} + m$ // update visit count
29:     $U_{S'} \leftarrow U_{S'} + v$ // update sum of reward
30:     $v_{S'} \leftarrow \frac{U_{S'}}{N_{S'}}$ // update value estimation
31:   **end for**
32: **end for**
33: Get value (reward) estimation of every solution $A_i \in D_{sol}$: $\{r_i = V_\phi(p_q, A_i)\}_{i=1}^{|D_{sol}|}$
34: $A^* = argmax_{A_i}(r_i)$ // Best-of-N
35: **Return** $A^*$
**Output:** $A^*$

---

second reward penalizes redundant characters (mainly meaningless characters after the end of the required program). In our experiments, we set $\omega_1 = 1e-3$ and $\omega_2 = 1e-6$. As for other settings, the question dataset is set to $Q_{train}$ within each self-training iteration. Taking into account time and resource constraints, we set $N = 30$, $\sigma_0 = 0.05$, $r_0 = 0.9$, $\beta = 0.5$ and $\alpha = 0.95$. When training with GRPO, the number of generations per group is set to $8$.

- **GRPO:** To validate the effectiveness of ReST on GRPO training, we involve naive GRPO as a baseline. For this baseline, the same reward function $R_{GRPO}$ is adopted for training. The number of generations per group is also set to $8$. In contrast, different from ReST-GRPO, questions in $Q_{train}$ are directly used for training. This means that each question in $Q_{train}$ will be used once within a single training iteration.

- **ReST-DPO:** We compare another self-train baseline ReST-DPO, which is similar to the ReST$^{EM}$ [24] method, except that the policy update algorithm is replaced from SFT to DPO [21]. For ReST-DPO, we also perform solution sampling with initial LLM policy on $Q_{train}$ within each self-train iteration, with the number of samples per question set to 30. Subsequently, the base reward function $R_{base}$ is deployed to obtain rewards for generated solutions, which are then used for the construction of preference pairs. Finally, we update the policy with the DPO objective, based on generated preference data.

$$R_{GRPO}(q, A) = R_{base}(q, A) + \omega_1 * \mathbb{I}(s \subseteq A) - \omega_2 * n\_redundant\_char(A),$$
where $s$ is an essential substring, and $\omega_1, \omega_2$ are weight parameters. (8)

### A.6 DETAILS FOR COMPARED DECODING AND VERIFICATION METHODS

We provide detailed descriptions and configurations of compared decoding and verification methods, i.e., VM-MCTS, ORM, PRM, and ORM-MCTS.

- **VM-MCTS:** For each base LLM policy, we train a corresponding VM based on Algorithm 2, using $Q_{train}$ as the question set. We set $T = 30$, $n = 5$, $c = 0.4$, $\epsilon = 0.1$ and $v_0 = 0$ when collecting value train data. During training, the VM $V_\phi$ is initialized by adding a classifier head to the base policy network. As for test-time decoding, we alter the parameter $c$ to 0.1 and determine the number of verification samples by changing the value of $T$, with other parameters remaining the same.

- **ORM:** We use the Skywork-Reward-Llama-3.1-8B-v0.2 ORM, which is trained on high-quality preference pairs sourced from publicly available data [15]. The output with the highest reward is returned.

- **PRM:** A Qwen2.5 PRM trained with an improved M-S method [39] is adopted to provide action-level reward. We use the minimum action-level reward for a single output as the final verification score. The output with the highest score is returned.

- **ORM-MCTS:** To assess the influence of MCTS on decoding accuracy, we include this baseline that also uses the Skywork ORM. We deploy a MCTS decoding algorithm similar to Algorithm 3, except that value estimation and final verification is based on the ORM.

### A.7 ADDITIONAL EXPERIMENTAL RESULTS

**Additional Results for Policy Training.** Different from training results in Table 1, we provide policy training results when using greedy decoding in test in Table 4. In addition, we also provide the training results of Llama-3.1-8B-Instruct and CodeQwen1.5-7B-Chat in Table 5. To ensure that the improvements achieved are beyond noise, we conduct four independent replicate self-train experiments under identical conditions mentioned in Section 3.2. In Table 6, we report self-train results with std-dev for Qwen3-8B on every benchmark. It can be observed that ReST-GRPO maintains clear margins ($> 5\%$ Avg., clearly not within random noise) over all other baselines and across iterations, indicating its significant improvement.

**Additional Results for Decoding and Verification.** To further demonstrate the difference in performance of compared decoding and verification methods, we provide detailed results of different base LLM policies when using ReST-RL and other methods on all benchmarks in Figure 4. We also display the average results of these methods for Llama-3.1-8B-Instruct and CodeQwen1.5-7B-Chat in Table 7. Moreover, to eliminate domain mismatch of other ORMs/PRMs, we train a Qwen3-8B based ORM (denoted as ORM (Trained)) on the same coding prompt dataset $Q_{train}$ using the identical unit-test reward as the VM. The test performance comparison is demonstrated in Table 8. The results show that even when using a dedicated ORM trained on the same dataset and employing MCTS to reinforce decoding ($54.9\%$ and $55.0\%$), it still falls short of ReST-RL's reasoning performance ($68.9\%$). This further solidifies the superiority of our approach.

**Ablation study on ReST-GRPO.** To clarify the practical role of ReST-GRPO's sampling method and hyperparameter selection, and to understand the contributions of filtering and partial sampling, we conduct ablation experiments. In Table 9, we present the ablation results for ReST-GRPO showing that our default setting (as illustrated in Appendix A.9, here represented by "$r_0 = 0.9$") for sampling

Table 4: Policy training results when using greedy decoding in test.

| Model | Training Method | HumanEval | HumanEval+ | MBPP | MBPP+ | APPS-500 | BCB | Avg. |
|---|---|---|---|---|---|---|---|---|
| Qwen3-8B | Base (0th iter.) | 0.848 | 0.787 | 0.717 | 0.619 | 0.116 | 0.43 | 0.508 |
| | *Below are results for sequential training iterations* | | | | | | | |
| | ReST-DPO (1st iter.) | 0.817 | 0.762 | 0.73 | 0.638 | 0.14 | 0.407 | 0.505 |
| | GRPO (1st iter.) | 0.829 | 0.793 | 0.743 | 0.64 | 0.347 | 0.445 | 0.574 |
| | **ReST-GRPO (1st iter.)** | 0.841 | 0.799 | 0.775 | 0.664 | 0.405 | 0.468 | 0.603 |
| | ReST-DPO (2nd iter.) | 0.823 | 0.774 | 0.741 | 0.656 | 0.153 | 0.416 | 0.517 |
| | GRPO (2nd iter.) | 0.829 | 0.787 | 0.757 | 0.656 | 0.412 | 0.448 | 0.594 |
| | **ReST-GRPO (2nd iter.)** | 0.878 | 0.829 | 0.794 | 0.698 | 0.552 | 0.482 | **0.658** |
| Qwen2.5-Coder-7B-Instruct | Base (0th iter.) | 0.909 | 0.841 | 0.828 | 0.714 | 0.302 | 0.419 | 0.592 |
| | *Below are results for sequential training iterations* | | | | | | | |
| | ReST-DPO (1st iter.) | 0.909 | 0.841 | 0.831 | 0.714 | 0.340 | 0.416 | 0.601 |
| | GRPO (1st iter.) | 0.896 | 0.835 | 0.833 | 0.714 | 0.384 | 0.425 | 0.612 |
| | **ReST-GRPO (1st iter.)** | 0.896 | 0.848 | 0.860 | 0.730 | 0.436 | 0.468 | 0.643 |
| | ReST-DPO (2nd iter.) | 0.909 | 0.848 | 0.825 | 0.714 | 0.356 | 0.422 | 0.607 |
| | GRPO (2nd iter.) | 0.890 | 0.829 | 0.844 | 0.728 | 0.392 | 0.428 | 0.616 |
| | **ReST-GRPO (2nd iter.)** | 0.909 | 0.860 | 0.854 | 0.725 | 0.416 | 0.492 | **0.646** |
| DS-Coder-6.7b-Instruct | Base (0th iter.) | 0.768 | 0.695 | 0.751 | 0.659 | 0.242 | 0.346 | 0.506 |
| | *Below are results for sequential training iterations* | | | | | | | |
| | ReST-DPO (1st iter.) | 0.780 | 0.707 | 0.746 | 0.661 | 0.245 | 0.348 | 0.510 |
| | GRPO (1st iter.) | 0.780 | 0.713 | 0.765 | 0.677 | 0.277 | 0.350 | 0.524 |
| | **ReST-GRPO (1st iter.)** | 0.787 | 0.707 | 0.767 | 0.685 | 0.287 | 0.366 | 0.532 |
| | ReST-DPO (2nd iter.) | 0.780 | 0.707 | 0.746 | 0.659 | 0.254 | 0.350 | 0.513 |
| | GRPO (2nd iter.) | 0.762 | 0.683 | 0.759 | 0.675 | 0.286 | 0.356 | 0.520 |
| | **ReST-GRPO (2nd iter.)** | 0.780 | 0.701 | 0.767 | 0.672 | 0.323 | 0.390 | **0.543** |
| OpenCI-DS-6.7B | Base (0th iter.) | 0.787 | 0.713 | 0.738 | 0.648 | 0.210 | 0.365 | 0.505 |
| | *Below are results for sequential training iterations* | | | | | | | |
| | ReST-DPO (1st iter.) | 0.774 | 0.713 | 0.733 | 0.643 | 0.207 | 0.373 | 0.503 |
| | GRPO (1st iter.) | 0.780 | 0.713 | 0.735 | 0.656 | 0.312 | 0.365 | 0.530 |
| | **ReST-GRPO (1st iter.)** | 0.780 | 0.713 | 0.738 | 0.656 | 0.324 | 0.373 | 0.535 |
| | ReST-DPO (2nd iter.) | 0.768 | 0.701 | 0.738 | 0.646 | 0.218 | 0.360 | 0.501 |
| | GRPO (2nd iter.) | 0.78 | 0.72 | 0.728 | 0.643 | 0.308 | 0.361 | 0.526 |
| | **ReST-GRPO (2nd iter.)** | 0.774 | 0.713 | 0.749 | 0.664 | 0.324 | 0.387 | **0.540** |

delivers the best variety of rewards, validating the core filtering and partial sampling methodology. From the perspective of reward variance (learning signal), we can see the significant importance of filtering operations ($0.104 \rightarrow 0.148$, related to $\sigma_0$ and $r_0$), but subsequent high-value trajectory selection and partial sampling are crucial as well ($0.148 \rightarrow 0.168$, related to $r_0$, $\alpha$ and $\beta$). Aside from reward variance, the proposed sampling strategy utilizes the high reward traces to automatically form valuable partial samples, this enables the LLM to thoroughly explore important action subspaces during optimization, which leads to observed further enhancements.

Table 5: Policy training results for Llama-3.1-8B-Instruct and CodeQwen1.5-7B-Chat.

| Model | Training Method | HumanEval | HumanEval+ | MBPP | MBPP+ | APPS-500 | BCB | Avg. |
|---|---|---|---|---|---|---|---|---|
| | Base | 0.671 | 0.591 | 0.693 | 0.569 | 0.175 | 0.301 | 0.435 |
| | Base (greedy) | 0.689 | 0.61 | 0.664 | 0.545 | 0.213 | 0.325 | 0.448 |
| | Below are results for the first training iteration | | | | | | | |
| | ReST-DPO | 0.628 | 0.579 | 0.669 | 0.569 | 0.209 | 0.313 | 0.436 |
| | GRPO | 0.652 | 0.573 | 0.675 | 0.566 | 0.271 | 0.324 | 0.457 |
| | **ReST-GRPO** | 0.677 | 0.610 | 0.735 | 0.616 | 0.248 | 0.383 | **0.488** |
| | ReST-DPO (greedy) | 0.677 | 0.616 | 0.701 | 0.574 | 0.259 | 0.321 | 0.466 |
| | GRPO (greedy) | 0.683 | 0.634 | 0.706 | 0.579 | 0.329 | 0.329 | 0.490 |
| Llama-3.1-8B-Instruct | **ReST-GRPO (greedy)** | 0.707 | 0.646 | 0.725 | 0.608 | 0.312 | 0.407 | **0.516** |
| | Below are results for the second training iteration | | | | | | | |
| | ReST-DPO | 0.671 | 0.598 | 0.672 | 0.556 | 0.249 | 0.299 | 0.449 |
| | GRPO | 0.671 | 0.598 | 0.701 | 0.585 | 0.284 | 0.341 | 0.476 |
| | **ReST-GRPO** | 0.665 | 0.598 | 0.693 | 0.587 | 0.235 | 0.418 | **0.481** |
| | ReST-DPO (greedy) | 0.677 | 0.616 | 0.698 | 0.574 | 0.272 | 0.325 | 0.470 |
| | GRPO (greedy) | 0.701 | 0.634 | 0.696 | 0.563 | 0.307 | 0.343 | 0.487 |
| | **ReST-GRPO (greedy)** | 0.720 | 0.659 | 0.741 | 0.624 | 0.342 | 0.425 | **0.535** |
| | Base | 0.805 | 0.732 | 0.81 | 0.677 | 0.148 | 0.348 | 0.502 |
| | Base (greedy) | 0.854 | 0.787 | 0.831 | 0.706 | 0.163 | 0.374 | 0.532 |
| | Below are results for the first training iteration | | | | | | | |
| | ReST-DPO | 0.823 | 0.774 | 0.759 | 0.659 | 0.167 | 0.353 | 0.507 |
| | GRPO | 0.829 | 0.750 | 0.788 | 0.653 | 0.244 | 0.365 | 0.530 |
| | **ReST-GRPO** | 0.854 | 0.793 | 0.804 | 0.690 | 0.27 | 0.389 | **0.557** |
| | ReST-DPO (greedy) | 0.854 | 0.787 | 0.817 | 0.696 | 0.185 | 0.373 | 0.534 |
| | GRPO (greedy) | 0.866 | 0.799 | 0.817 | 0.698 | 0.272 | 0.401 | 0.566 |
| CodeQwen1.5-7B-Chat | **ReST-GRPO (greedy)** | 0.890 | 0.835 | 0.831 | 0.714 | 0.294 | 0.447 | **0.594** |
| | Below are results for the second training iteration | | | | | | | |
| | ReST-DPO | 0.841 | 0.780 | 0.799 | 0.672 | 0.176 | 0.339 | 0.515 |
| | GRPO | 0.878 | 0.823 | 0.815 | 0.672 | 0.255 | 0.375 | 0.556 |
| | **ReST-GRPO** | 0.878 | 0.816 | 0.828 | 0.706 | 0.278 | 0.425 | **0.579** |
| | ReST-DPO (greedy) | 0.860 | 0.787 | 0.836 | 0.709 | 0.211 | 0.362 | 0.542 |
| | GRPO (greedy) | 0.866 | 0.799 | 0.828 | 0.701 | 0.296 | 0.401 | 0.574 |
| | **ReST-GRPO (greedy)** | 0.902 | 0.835 | 0.836 | 0.725 | 0.308 | 0.479 | **0.609** |

Table 6: Reinforce self training results for Qwen3-8B with standard deviation.

| Model | Training Method | HumanEval | HumanEval+ | MBPP | MBPP+ | APPS-500 | BCB | Avg. |
|---|---|---|---|---|---|---|---|---|
| | Base (0th iter.) | 0.829±0.018 | 0.780±0.019 | 0.717±0.003 | 0.622±0.002 | 0.118±0.003 | 0.418±0.007 | 0.503±0.008 |
| | Below are results for sequential training iterations | | | | | | | |
| | ReST-DPO (1st iter.) | 0.854±0.013 | 0.799±0.006 | 0.730±0.005 | 0.627±0.011 | 0.152±0.015 | 0.434±0.010 | 0.523±0.011 |
| | GRPO (1st iter.) | 0.799±0.018 | 0.750±0.030 | 0.754±0.016 | 0.651±0.013 | 0.346±0.006 | 0.439±0.004 | 0.566±0.012 |
| Qwen3-8B | **ReST-GRPO (1st iter.)** | 0.872±0.012 | 0.817±0.006 | 0.780±0.005 | 0.672±0.003 | 0.377±0.008 | 0.469±0.013 | 0.604±0.009 |
| | ReST-DPO (2nd iter.) | 0.841±0.006 | 0.811±0.012 | 0.741±0.006 | 0.653±0.007 | 0.153±0.011 | 0.429±0.007 | 0.526±0.008 |
| | GRPO (2nd iter.) | 0.829±0.006 | 0.787±0.006 | 0.757±0.006 | 0.667±0.014 | 0.403±0.007 | 0.436±0.004 | 0.590±0.007 |
| | **ReST-GRPO (2nd iter.)** | 0.860±0.012 | 0.805±0.012 | 0.802±0.006 | 0.690±0.010 | 0.565±0.022 | 0.476±0.007 | **0.655±0.012** |

Table 7: Average results of ReST-RL and different verification methods on all benchmarks for Llama-3.1-8B-Instruct and CodeQwen1.5-7B-Chat.

| Method | Llama-3.1-8B-Instruct | CodeQwen1.5-7B-Chat |
|---|---|---|
| Base | 0.435 | 0.502 |
| ORM | 0.480 | 0.542 |
| PRM | 0.466 | 0.526 |
| ORM-MCTS | 0.481 | 0.545 |
| VM-MCTS | 0.519 | 0.599 |
| ReST-RL | **0.556** | **0.616** |

Table 8: Detailed test verification results for Qwen3-8B. To ensure domain parity, we include two extra baselines that adopt an ORM trained on the same dataset $Q_{train}$.

| Model | Method | HumanEval | HumanEval+ | MBPP | MBPP+ | APPS-500 | BCB | Avg. |
|---|---|---|---|---|---|---|---|---|
| Qwen3-8B | Base | 0.829 | 0.780 | 0.717 | 0.622 | 0.118 | 0.418 | 0.503 |
| | ORM (Skywork) | 0.841 | 0.787 | 0.746 | 0.661 | 0.165 | 0.441 | 0.531 |
| | PRM (Qwen) | 0.860 | 0.817 | 0.714 | 0.614 | 0.138 | 0.423 | 0.516 |
| | ORM (Skywork) + MCTS | 0.854 | 0.805 | 0.746 | 0.661 | 0.193 | 0.424 | 0.538 |
| | ORM (Trained) | 0.854 | 0.823 | 0.746 | 0.643 | 0.214 | 0.449 | 0.549 |
| | ORM (Trained) + MCTS | 0.860 | 0.826 | 0.743 | 0.648 | 0.221 | 0.441 | 0.550 |
| | VM-MCTS | 0.866 | 0.829 | 0.775 | 0.675 | 0.415 | 0.471 | 0.615 |
| | **ReST-RL** | 0.866 | 0.829 | 0.817 | 0.704 | 0.642 | 0.506 | **0.689** |

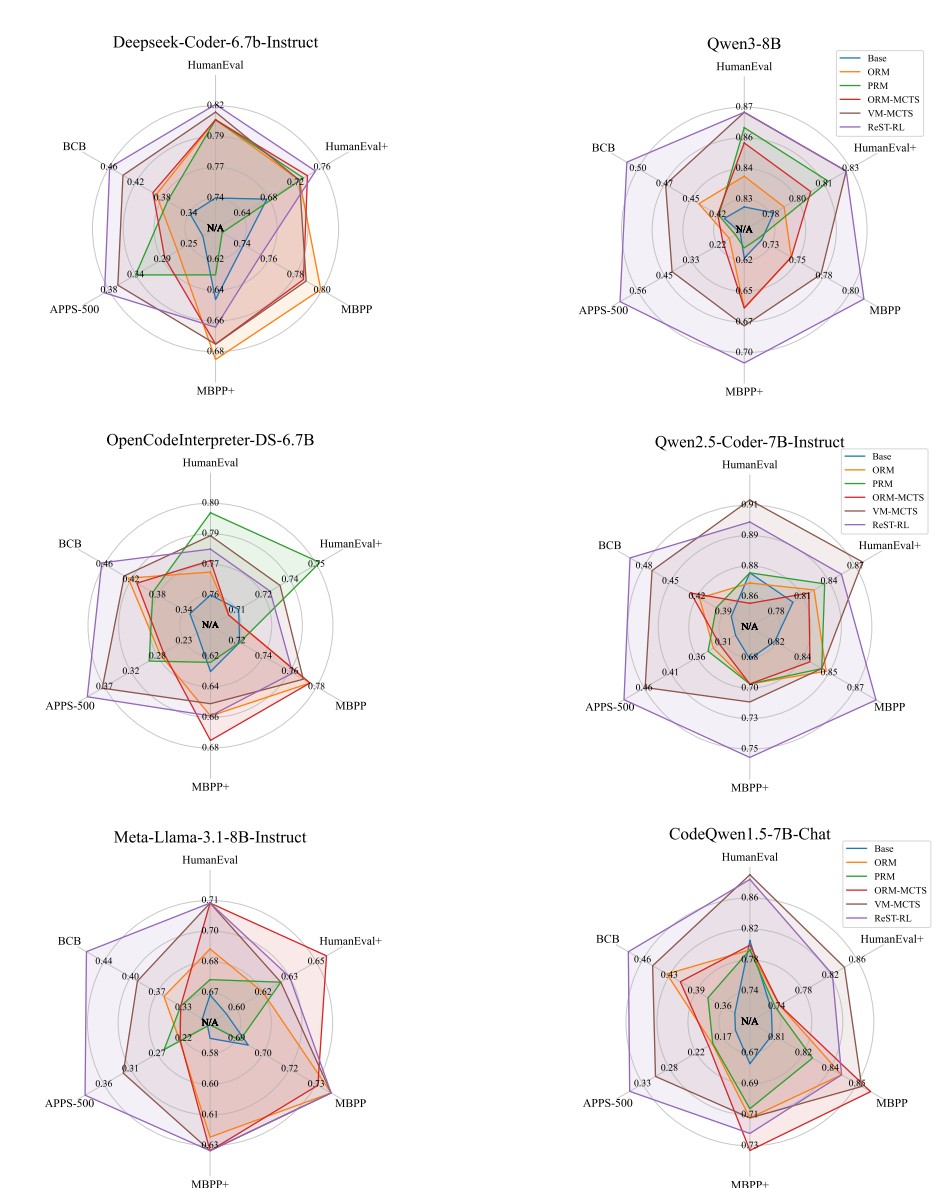

Figure 4: Performance of different base LLM policies when using ReST-RL and verification methods on all benchmarks. All verification is based on 100 samples, with the sampling temperature set to 0.7.

Table 9: Ablation results for the train data sampling method of ReST-GRPO. The mean and median reward standard deviation of train samples acquired through different methods are recorded.

| Sampling Method | $\sigma_0 = 0, \beta = 0$ | $\alpha \to 0(0.01)$ | $r_0 = 0.1$ | $r_0 = 0.5$ | $r_0 = 0.9$ |
|---|---|---|---|---|---|
| Mean SD | 0.104 | 0.148 | 0.157 | 0.163 | **0.168** |
| Median SD | 0.000 | 0.120 | 0.111 | 0.124 | **0.130** |

Table 10: Average token usage per prompt for CodeQwen. We compare ORM+Best-of-N with our VM-MCTS method, showing the number of decoded tokens under various sampling sizes.

| Model | Method | N=5 | N=10 | N=20 | N=50 | N=100 |
|---|---|---|---|---|---|---|
| CodeQwen | Best-of-N | 1290 | 2566 | 5137 | 12800 | 25564 |
| | VM-MCTS | 1287 | 2544 | 4962 | 12055 | 22427 |

### A.8 MATHEMATICAL PROOF FOR VALUE ESTIMATION

In this part we prove that our proposed value-based MC rollout tends to have more precise estimation of partial state's value, in terms of variance. Suppose we are estimating the value of partial state $S_i = (q, a_{1,2,\ldots,i})$ for a fixed policy $\pi$. We are allowed to use two verifier models, a value model $V_\phi$ and a reward model $R_\lambda$ (which is allowed to be identical to $V_\phi$), while the actual reward function $R$ and value function $V$ are unknown (since it's test time). As a reasonable approximation, we assume that the two models can be regarded as two unbiased estimator variables that have the same variance when predicting, along with some conditions of independence:

$$
\begin{aligned}
V_\phi(S) &= V(S) + \epsilon_V, \\
R_\lambda(S_{end}) &= R(S_{end}) + \epsilon_R, \\
s.t.\ \mathbb{D}[\epsilon_V] &= \mathbb{D}[\epsilon_R], \mathbb{E}[\epsilon_V] = \mathbb{E}[\epsilon_R] = 0, \\
\epsilon_V &\text{ is independent of } S, \\
\epsilon_R &\text{ is independent of } S_{end}.
\end{aligned} \tag{9}
$$

Now, we consider two approaches for value estimation, with a limit of $n$ simulations. For value-based MC rollout, it does simulation with the random variable $S_{i+1}$, i.e. the state reached by $\pi$ performing a single action. It estimates $V(S_i)$ by:

$$
\hat{V}_v(S_i) = \frac{1}{n} \sum_{j=1}^{n} V_\phi(S_{i+1}^{(j)}), \tag{10}
$$

$$
s.t.\ S_{i+1}^{(1)}, S_{i+1}^{(2)}, \ldots, S_{i+1}^{(n)} \overset{\text{i.i.d.}}{\sim} \pi(S_{i+1}|S_i).
$$

On the other hand, regular complete rollout does simulation with the random variable $S_{end}$, i.e. the end state reached after a complete trace is generated. It estimates $V(S_i)$ by:

$$
\hat{V}_r(S_i) = \frac{1}{n} \sum_{j=1}^{n} R_\lambda(S_{end}^{(j)}), \tag{11}
$$

$$
s.t.\ S_{end}^{(1)}, S_{end}^{(2)}, \ldots, S_{end}^{(n)} \overset{\text{i.i.d.}}{\sim} \pi(S_{end}|S_i).
$$

We now state the proposition to be proved:

$$
\mathbb{E}[\hat{V}_r(S_i)] = \mathbb{E}[\hat{V}_v(S_i)] = V(S_i), \tag{12}
$$

and

$$
\mathbb{D}[\hat{V}_r(S_i)] \geq \mathbb{D}[\hat{V}_v(S_i)]. \tag{13}
$$

If the statements are true, we can conclude that $\hat{V}_v(S_i)$ is a better estimator than $\hat{V}_r(S_i)$ due to its smaller variance, which justifies our original claim. Now, we provide the proof for Eq. (12).

*Proof.* First, we notice that by Eq. (9):

$$
\begin{aligned}
\hat{V}_r(S_i) &= \frac{1}{n} \sum_{j=1}^{n} R_\lambda(S_{end}^{(j)}) \\
&= \frac{1}{n} \sum_{j=1}^{n} (R(S_{end}^{(j)}) + \epsilon_R) \\
&= \epsilon_R + \frac{1}{n} \sum_{j=1}^{n} R(S_{end}^{(j)})
\end{aligned}
$$

Substituting the above equation into the expression of expectation, we derive that:

$$\mathbb{E}[\hat{V}_r(S_i)] = \mathbb{E}[\epsilon_R + \frac{1}{n}\sum_{j=1}^{n} R(S_{end}^{(j)})]$$

$$= \mathbb{E}[\epsilon_R] + \frac{1}{n}\sum_{j=1}^{n} \mathbb{E}[R(S_{end}^{(j)})]$$

$$= \frac{1}{n}\sum_{j=1}^{n} \mathbb{E}[R(S_{end})|S_i]$$

$$= \mathbb{E}[R(S_{end})|S_i]$$

$$= V(S_i)$$

Similarly, we can know that:

$$\hat{V}_v(S_i) = \frac{1}{n}\sum_{j=1}^{n} V_\phi(S_{i+1}^{(j)})$$

$$= \frac{1}{n}\sum_{j=1}^{n}(V(S_{i+1}^{(j)}) + \epsilon_V)$$

$$= \epsilon_V + \frac{1}{n}\sum_{j=1}^{n} V(S_{i+1}^{(j)})$$

Thus, we can finally derive the desired properties:

$$\mathbb{E}[\hat{V}_v(S_i)] = \mathbb{E}[\epsilon_V + \frac{1}{n}\sum_{j=1}^{n} V(S_{i+1}^{(j)})]$$

$$= \mathbb{E}[\epsilon_V] + \frac{1}{n}\sum_{j=1}^{n} \mathbb{E}[V(S_{i+1}^{(j)})]$$

$$= \frac{1}{n}\sum_{j=1}^{n} \mathbb{E}[V(S_{i+1})|S_i]$$

$$= \mathbb{E}[V(S_{i+1})|S_i]$$

$$= V(S_i)$$

$\square$

Next, we provide proof for Eq. (13) as follows.

*Proof.* Using the conclusions made by the proof above, we can derive the required property:

$$\mathbb{D}[\hat{V}_r(S_i)] \geq \mathbb{D}[\hat{V}_v(S_i)]$$

$$\Longleftarrow \mathbb{D}[\epsilon_R + \frac{1}{n}\sum_{j=1}^{n} R(S_{end}^{(j)})] \geq \mathbb{D}[\epsilon_V + \frac{1}{n}\sum_{j=1}^{n} V(S_{i+1}^{(j)})]$$

$$\Longleftarrow \mathbb{D}[\epsilon_R] + \mathbb{D}[\frac{1}{n}\sum_{j=1}^{n} R(S_{end}^{(j)})] \geq \mathbb{D}[\epsilon_V] + \mathbb{D}[\frac{1}{n}\sum_{j=1}^{n} V(S_{i+1}^{(j)})]$$

$$\Longleftarrow \frac{1}{n^2}\mathbb{D}[\sum_{j=1}^{n} R(S_{end}^{(j)})] \geq \frac{1}{n^2}\mathbb{D}[\sum_{j=1}^{n} V(S_{i+1}^{(j)})]$$

$$\Longleftarrow \sum_{j=1}^{n}\mathbb{D}[R(S_{end}^{(j)})] \geq \sum_{j=1}^{n}\mathbb{D}[V(S_{i+1}^{(j)})]$$

$$\Longleftarrow \mathbb{D}[R(S_{end})|S_i] \geq \mathbb{D}[V(S_{i+1})|S_i]$$

$$\Longleftarrow \mathbb{E}[(R(S_{end}))^2|S_i] - (\mathbb{E}[R(S_{end})|S_i])^2 \geq \mathbb{E}[(V(S_{i+1}))^2|S_i] - (\mathbb{E}[V(S_{i+1})|S_i])^2$$

$$\Longleftarrow \mathbb{E}[(R(S_{end}))^2|S_i] \geq \mathbb{E}_{S_{i+1}\sim\pi(S_{i+1}|S_i)}[(V(S_{i+1}))^2|S_i]$$

$$\Longleftarrow \mathbb{E}_{S_{i+1}\sim\pi(S_{i+1}|S_i)}[\mathbb{E}[(R(S_{end}))^2|S_{i+1}]|S_i] \geq \mathbb{E}_{S_{i+1}\sim\pi(S_{i+1}|S_i)}[(\mathbb{E}[R(S_{end})|S_{i+1}])^2|S_i]$$

$$\Longleftarrow \mathbb{E}_{S_{i+1}\sim\pi(S_{i+1}|S_i)}[\mathbb{E}[(R(S_{end}))^2|S_{i+1}]] \geq \mathbb{E}_{S_{i+1}\sim\pi(S_{i+1}|S_i)}[(\mathbb{E}[R(S_{end})|S_{i+1}])^2]$$

$$\Longleftarrow \mathbb{E}_{S_{i+1}\sim\pi(S_{i+1}|S_i)}[\mathbb{E}[(R(S_{end}))^2|S_{i+1}] - (\mathbb{E}[R(S_{end})|S_{i+1}])^2] \geq 0$$

$$\Longleftarrow \mathbb{E}_{S_{i+1}\sim\pi(S_{i+1}|S_i)}[\mathbb{D}[R(S_{end})|S_{i+1}]] \geq 0$$

$$\Longleftarrow \mathbb{D}[R(S_{end})|S_{i+1}] \geq 0$$

$$\square$$

## A.9 TABLE OF HYPERPARAMETERS

Table 11: Main hyperparameters and default values for the ReST-GRPO method.

| Hyperparameter | $N$ | $\sigma_0$ | $r_0$ | $\alpha$ | $\beta$ |
|---|---|---|---|---|---|
| Meaning | Number of solution samples | SD threshold | Reward threshold | Sampling exponent factor | Sample ratio |
| Default Value | 30 | 0.05 | 0.9 | 0.95 | 0.5 |

Table 12: Main hyperparameters and default values for the VM-MCTS method.

| Hyperparameter | $T$ | $n$ | $c$ | $\epsilon$ |
|---|---|---|---|---|
| Meaning | Search iterations | Number of branches expanded | Exploration constant | Constant avoiding zero-division in UCT |
| Default Value | 30 | 5 | 0.4 | 0.1 |

