# OpenReview forum: "ReST-RL: Reinforcing LLM Reasoning through Self-Training and Value-Guided Decoding"
_ICLR.cc/2026/Conference — Submitted to ICLR 2026_

### Official Review · Reviewer_zamV · 2025-10-25

**Soundness:** 2
**Presentation:** 2
**Contribution:** 2
**Rating:** 2
**Confidence:** 4

**Summary:**

This paper introduces ReST-RL, a unified RL paradigm that enhances LLM reasoning for code generation. It operates in two stages: ReST-GRPO, a training algorithm that increases reward variance via self-training, and VM-MCTS, a decoding method that uses a Value Model to guide the LLM. The main contributions are as follows:
1. It proposes the ReST-RL paradigm, which effectively integrates the advantages of offline self-training, online policy optimization, and value-guided decoding to achieve a balance between effectiveness, efficiency, and data cost.
2. It introduces two key technical components: the ReST-GRPO algorithm for more effective policy training and the VM-MCTS method for enhanced decoding, both of which are shown to outperform strong baselines.
3. It demonstrates significant performance gains over existing methods across multiple LLMs and challenging code benchmarks , showcasing the approach's power to strengthen LLM reasoning.

**Strengths:**

1. The work shows strong originality by creatively unifying offline self-training, online RL, and value-guided tree search into a novel, two-stage paradigm. This synthesis effectively addresses key limitations of prior methods, namely low reward variance and high annotation costs for process supervision.
2. The paper's quality is demonstrated through extensive, rigorous experiments across multiple code benchmarks and base LLMs. Results consistently show significant improvements over strong baselines, with convincing ablation studies and efficiency analysis, providing robust validation for the proposed method.
3. The presentation is exceptionally clear, with a well-motivated narrative and detailed algorithms. The work is significant for delivering a practical framework that balances performance, efficiency, and data cost, offering a valuable blueprint for enhancing LLM reasoning in complex domains like code generation.

**Weaknesses:**

1. The paper omits comparisons with recent, highly relevant state-of-the-art methods. Specifically, there is no comparison with R1-style process-based reinforcement learning or the closely related PSPO framework, which also combines process reward with policy optimization. Demonstrating superiority over these established strong baselines is crucial to fully validate the claimed contributions.
2. While the method claims efficiency, it lacks a detailed analysis of the computational overhead. The combined cost of iterative ReST-GRPO training plus the VM-MCTS decoding pipeline is likely substantial. A breakdown of training time, inference latency, and GPU memory footprint compared to baselines is needed to assess the practical trade-offs of the proposed approach.
3. The Value Model is trained and tested on distributions from the same policy and dataset. Its performance and reliability when guiding the LLM on out-of-domain problems or after significant policy updates are not investigated. This raises questions about the long-term applicability and robustness of the VM-MCTS component without frequent, costly retraining.
4. The reliance on a purely rule-based, test-case-passing reward, while avoiding reward hacking, ignores richer signals of code quality. This may limit the model's ability to learn truly generalizable coding principles. Incorporating a learned reward model, even with its risks, could be a necessary evolution for more advanced reasoning.

**Questions:**

1. Could you provide a concrete breakdown of the computational overhead for both ReST-GRPO training and VM-MCTS decoding compared to key baselines? This is crucial for assessing the practical deployability of your method.
2. Why were comparisons with strong, relevant baselines like R1 or PSPO omitted? Including these would more convincingly demonstrate the superiority of your proposed approach against the current state-of-the-art.
3. How does the performance of your Value Model degrade when the underlying policy is updated or when applied to out-of-domain problems? An analysis of its robustness to such distribution shifts is needed to assess long-term viability.

---

> ### Author Response · Authors · 2025-11-21
> **Response to Reviewer zamV (1/2)**
>
> We appreciate your recognition of the framework's originality, experimental rigor, and clear presentation. Below we provide the requested details.
>
> **W1&Q2: Missing comparisons with R1 / PSPO.**
> Thank you for highlighting the importance of these recent related works. However, we must point out that some approaches, exemplified by Deepseek R1 (https://arxiv.org/pdf/2501.12948), primarily rely on verifiable outcome rewards rather than process rewards for reinforcement training and employ standard GRPO algorithms (which is one of our baseline setting). According to the R1 paper, they failed to successfully train a scalable PRM. They also pointed out the issues with traditional PRM: difficulty in judging the correctness of intermediate steps, as well as the problem of reward hacking. As for the PSPO method, it proposes a workflow spanning from PRM training to policy optimization to address logical errors and redundant outputs caused by the lack of process supervision in LLM mathematical reasoning. It emphasizes the importance of nonlinear rewards based on output accuracy and length in process supervision, achieving improvements on mathematical reasoning tasks. However, PSPO essentially proposes a variant of PRM rather than the VM defined in our paper. In Table 2 of the main paper, we have compared VM with PRM and demonstrated the superiority of VM. Moreover, labeling and training costs for these PRMs are extremely high (PSPO requires substantial human annotation on process labels). Our sampling design for VM-MCTS and the use of test-case-based reward specifically address this issue. Furthermore, PSPO primarily targets mathematical reasoning, which differs from our main objective (code reasoning). Therefore, we did not consider it when selecting baselines.
>
> **W2&Q1: Lack of computational cost breakdown.**
>
> Here we present a cost breakdown from the training and decoding perspective to defend our claim. Table 1 shows the GPU-hours required for every 1k training samples for ReST-GRPO and naive GRPO. Due to its pre-train filtering and partial sampling design, ReST-GRPO attains an **approximately 20% reduction on training time, while achieving better training results** (Figure 3(a) in our paper). This confirms our claim regarding efficiency. Note that the pre-train sampling process of ReST-GRPO only equals to 15-20% of the training time, making the cost entirely acceptable.
>
> In Table 2 we display the decoding cost of VM-MCTS compared to conventional Best-of-N, where `N` is the total number of sampled traces. Combined with Figure 3(b), we can observe that **VM-MCTS significantly outperforms Best-of-N across all sampling sizes while requiring fewer sampled tokens**, owing to its ability to perform rollouts from partial samples.
>
> *Table 1: Training GPU-hour accounting (Llama-3-8B, we have supplemented these results in Figure 3 in the main paper).*
>
> | GPU Hours / k samples | 1   |  2  |  3  |  4  |  5  |  6  |  7  |  8  |  9  |  10  |
> | --------------------- | --- | :-: | :-: | :-: | :--: | :--: | :--: | :--: | :--: | :--: |
> | GRPO                  | 209 | 415 | 612 | 819 | 1035 | 1251 | 1454 | 1654 | 1865 | 2081 |
> | ReST-GRPO             | 163 | 325 | 482 | 640 | 818 | 990 | 1168 | 1344 | 1515 | 1672 |
>
> *Table 2: Average decoding computation cost (k tokens per prompt, CodeQwen, we have supplemented these results in Figure 3 in the main paper).*
>
> | Avg. Tokens (k) / prompt | N=5 | N=10 | N=20 | N=50 | N=100 |
> | ------------------------ | :--: | :--: | :--: | :---: | :---: |
> |  Best-of-N                 | 1.29 | 2.57 | 5.14 | 12.80 | 25.56 |
> | VM-MCTS                | 1.28 | 2.54 | 4.96 | 12.06 | 22.43 |

---

> > ### Author Response · Authors · 2025-11-21
> > **Response to Reviewer zamV (2/2)**
> >
> > **W3&Q3: VM robustness under policy shift / OOD prompts.**
> > It is indeed crucial to confirm the generalization capability of the ReST-RL method and the VM on out-of-domain data. In our original paper, we included Humaneval(+) and MBPP(+) as test benchmarks, whose relevant data never appeared during training. The experimental results demonstrate that the ReST-RL method and the VM performs well on these test datasets (Table 2 and Figure 4 in the paper). Furthermore, Table 3 below examines its generalizability on math reasoning tasks (MATH, Omni-MATH). We can observe that even after undergoing two full rounds of policy reinforcement learning, the initial VM retains strong applicability. Although its score falls short of the similarly updated VM, the performance loss is relatively minor (approximately 1%). **These results demonstrate that ReST-RL and its VM have strong transferability and generalizability across unseen reasoning domains and policy checkpoints**. Note that this competitive performance enhancement on MATH/Omni-MATH is achieved **without any math-specific tuning**, confirming that ReST-RL extends beyond coding.
> >
> > *Table 3: Generalization of the ReST-RL approach to out-of-domain prompts (Qwen3-8B, we have supplemented these results in Section 3.4 in the main paper).*
> >
> > | Method                                 |    APPS-500    |       BCB       |      MATH      |    Omni-MATH    |
> > | -------------------------------------- | :-------------: | :-------------: | :-------------: | :-------------: |
> > | Base (0th iter.)                       |      0.118      |      0.418      |      0.780      |      0.234      |
> > | Base (0th iter.) + VM (0th iter.)      |      0.415      |      0.471      |      0.828      |      0.238      |
> > | ReST-GRPO (2nd iter.) + VM (0th iter.) |      0.630      |      0.496      |      0.862      |      0.246      |
> > | ReST-GRPO (2nd iter.) + VM (2th iter.) | **0.642** | **0.506** | **0.872** | **0.256** |
> >
> > **W4: Limitation of rule-based rewards.**
> > Your suggestion to refine reward signals using trained reward models is insightful. In fact, our reinforcement framework does not impose any specific form on the reward function. Given our focus on code reasoning and considerations of method scalability (mainstream reinforcement paradigms like R1 indicate that using easily verifiable RLVRs offers high scalability), we adopted test-case-based rewards in our experiments rather than reward models requiring additional data collection and training. Our results demonstrate that this simple reward is sufficient to achieve significant improvements. That said, we do not dismiss the possibility that more refined reward models could yield further gains.
> >
> > Overall, we hope these clarifications demonstrate that ReST-RL is practical, powerful, robust and extensible. Thank you again for the insightful feedback. If you feel our response has addressed some of your concerns, we sincerely hope you will adjust your rating. Thank you!

---

> ### Author Response · Authors · 2025-11-28
> **Reminder**
>
> **Dear Reviewer zamV,**
>
> Thank you again for the time and effort you have devoted to reviewing our submission. We have provided detailed responses and additional clarifications to all reviewer comments. As the rebuttal period is approaching its final deadline, we would kindly like to remind you to check whether any further feedback or follow-up questions are needed from our side. We sincerely appreciate your consideration and would be grateful for any updates on the scores at your convenience.
>
> Thank you very much for your time.

---

### Official Review · Reviewer_SYgK · 2025-10-29

**Soundness:** 2
**Presentation:** 2
**Contribution:** 2
**Rating:** 2
**Confidence:** 3

**Summary:**

The paper proposes a two-stage framework to improve code-reasoning LLMs: (1) ReST-GRPO, which filters prompts by reward variance and bootstraps training with partial-trace starting states before GRPO updates; and (2) VM-MCTS, which trains a value model with MCTS-collected targets and uses it for value-guided search/verification at test time. Experiments on HumanEval/MBPP/APPS-500/BCB suggest consistent gains over naive GRPO/ReST-DPO and over ORM/PRM-based verification, with additional curves for train-step efficiency and budgeted verification.

**Strengths:**

The paper investigates an interesting area and proposes a combination of existing methods. The proposed partial-state solution is particularly compelling and intuitively sound. Experimental results show that ReST-RL and VM-MCTS achieve superior performance across various coding benchmarks.

**Weaknesses:**

1. The paper proposes to combine existing methods, ReST, an offline self-training paradigm with GRPO, an on-policy RL algorithm (stated by the authors at line 261). I have concerns with the distribution shifting issue in the loss update. The original ReST investigates multiple offline RL methods, the ReST-MCTS* and ReST$_{EM}$ are using SFT. Given the authors' design of the partial state completion, the training data distribution is shifted comparing to the test prompt if they use common GRPO objective as they stated at line 262. I acknowledge that p.m.f at equation (3) might partially mitigate it as it gives probability that the data fall back to original GRPO. I think it worths to investigate and discuss why ReST-GRPO performs so well and how is the distribution shifting coped. It would be great to add ablation experiments for different alpha in p.m.f might give more insights.
2. VM-MCTS and ReST-GRPO seems to be distinct components. VM-MCTS can help test time computation is not novel.  In PPO-MCTS, the value function is a byproduct of the PPO training while GRPO does not have a value model. Using a smaller value network to help the LLM in the test time has been proposed by "Reasoning with Language Model is Planning with World Model" before and the comparing between VM-MCTS and ORM and PRM with Best-of-N seems ignoring other strong tree search baselines.

Other minor issues:
1.Over-claiming the general reasoning benefits in the introduction but all experiments are code benchmark. I think either be specific about the benefit of reasoning capability in coding or add experiments with math benchmark will be more convincing.
2. The caption of the figure and table is not stand along explainable. For instance, in Table 1, what does the 1st iter and 2nd iter mean. I think it would add clarity by introduce it in the caption.

**Questions:**

1. Does the authors use any way to address the distribution shifting in the training set? If not, why?
2. What is the purpose of VM-MCTS? Is it just to show that it can improve the test time performance? In experiments section, does the experiments with ReST-GRPO apply the VM-MCTS in the test time?

---

> ### Author Response · Authors · 2025-11-21
> **Response to Reviewer SYgK (1/2)**
>
> Thank you for highlighting the novelty of partial-state completion and the strength of our experimental evidence. We address your questions below.
>
> **W1&Q1: Distribution shift due to partial-state completion.**
>
> In fact, the question you raise is precisely the reason and core of ReST-GRPO's design. As discussed in Section 2.2, traditional offline learning methods (DPO, ReST-EM, ReST-MCTS) are constrained by existing static trajectories and cannot fully explore the action space. Conversely, online learning methods (GRPO, PPO) often perform extensive ineffective exploration due to the lack of trajectory guidance, limiting training effectiveness. **ReST-GRPO was designed to integrate the sampling patterns of offline learning into online learning. This actively alters the distribution of training samples, enabling the policy model to conduct efficient exploration within selected high-value subspaces.** This is similar to learning a new sport: compared to directly practicing through full games, it's often more effective to first practice specific techniques within the sport. Consequently, ReST-GRPO achieves a breakthrough beyond existing reinforcement learning paradigms. Considering that GRPO's model optimization relies on reward variances across trajectories, we designed pre-sampling, filtering, and partial sampling methods for training samples. This approach seeks initial training states with high sample reward variance to identify high-value search subspaces.
>
> In Table 1 below, we present an ablation for ReST-GRPO showing that our default setting (as illustrated in Appendix A.5 and A.6, here represented by `r₀=0.9`) for sampling delivers the best variety of rewards, validating the **filtering and partial sampling** methodology. From the perspective of reward variance (learning signal), we can see the significant importance of filtering operations (0.104 → 0.148), but subsequent high-value trajectory selection and partial sampling are crucial as well (0.148 → 0.168).
>
> In summary, ReST-GRPO's proactive modification of the sampling distribution is not a problem to be solved, but rather a powerful methodology for enhancing the effectiveness of reinforcement learning. We believe it enables the LLM to **thoroughly explore important action subspaces** during optimization, which leads to observed further enhancements.
>
> *Table 1: Ablation on hyperparameters for the ReST-GRPO algorithm from the perspective of reward standard deviation (Qwen3-8B, we have supplemented it in Appendix A.7 in the main paper).*
>
> | Sampling Method | σ₀=0, β=0 | α → 0(0.01) | r₀=0.1 | r₀=0.5 | r₀=0.9         |
> | :-------------: | :----------: | :-----------: | :-----: | :-----: | --------------- |
> |     Mean SD     |    0.104    |     0.148     |  0.157  |  0.163  | **0.168** |
> |    Median SD    |    0.000    |     0.120     |  0.111  |  0.124  | **0.130** |
>
> **W2&Q2: Novelty of VM-MCTS and comparison with tree-search methods.**
>
> Experiments specifically targeting ReST-GRPO **do not employ VM-MCTS** (e.g., Table 1 in the paper). It is important to note that **VM-MCTS represents a novel approach (it's not conventional VM + MCTS), differing from traditional VMs in its definition, training, and utilization methods**. Specifically, our definition of the VM encompasses two aspects: for intermediate states, the VM predicts the expected reward value; whereas for terminal (end) states, it predicts a deterministic reward value (this reward is validated upon reaching the terminal state, not prior to reaching it). This design endows our VM with characteristics of both ORM and PRM (or traditional VMs), enabling it to provide richer and more precise validation signals. Traditional VMs are obtained through online learning, but co-evolving with policy models introduces convergence risks. Deploying multiple models for simultaneous training also imposes extremely high demands on computational resources. Methods like PPO-MCTS that obtain VMs this way suffer from the aforementioned issues and, during decoding, can only perform token-level searches. For complex tasks, the computational cost is nearly prohibitive (hence their limited adoption). Compared to these approaches, the VM-MCTS method offers the following significant advantages:
>
> 1. Decoupled policy and value training reduces training resource constraints and promotes model convergence.
> 2. Adapt to a broader range of states and action spaces for value estimation (e.g., lines or paragraphs).
> 3. Provides estimates of intermediate and terminal states, serving as both a steering signal for MCTS and a result validator.
> 4. Unlike standard MCTS, VM-MCTS directly employ the trained VM to perform partial state estimation, thereby reducing the variance in rollout estimates (Appendix A.8).

---

> > ### Author Response · Authors · 2025-11-21
> > **Response to Reviewer SYgK (2/2)**
> >
> > In the paper, we did not consider only the Best-of-N baseline, but also a strong tree search baseline ORM+MCTS. Methods like "Reasoning with Language Model is Planning with World Model" employs the LLM itself as the reward evaluator. However, its scoring process relies on specific prompts and exhibits poor cross-task scalability. Besides, ReST-MCTS and "Let's verify step by step" also indicate that ORM/PRM trained with external explicit rewards outperform LLM self-scoring. Therefore, we selected the ORM+MCTS approach as our tree search baseline.
> >
> > To comprehensively validate VM-MCTS, we evaluated several different strong open-source reward models, including ORM (Skywork), PRM (Qwen), and an ORM (Trained) specifically trained on the same coding dataset `Q_{train}` using the identical unit-test reward as the VM, enhanced with tree search via standard MCTS. The test performance comparison is shown in Table 2 below (and also in Table 2 in the paper). The results show that **even when using a dedicated ORM trained on the same dataset and employing MCTS to reinforce decoding (54.9% and 55.0%), it still falls short of VM-MCTS and ReST-RL's reasoning performance (61.5% and 68.9%)**. This further solidifies the superiority of our approach.
> >
> > **Overall, although ReST-GRPO and VM-MCTS are two distinct reinforcement learning stages, their designs are closely intertwined**. Through meticulous exploration of intermediate states, both stages incorporate the strengths of multiple methodologies, decoupling complex policy and value training, while significantly enhancing the reasoning capabilities of LLMs. Extensive experiments demonstrate the effectiveness of the ReST-GRPO and VM-MCTS designs.
> >
> > *Table 2: Test performance of ReST-RL and other enhanced decoding baselines (Qwen3-8B, we have supplemented it in Appendix A.7 in the main paper).*
> >
> > | Method               | HumanEval | HumanEval+ | MBPP | MBPP+ | APPS-500 |  BCB  |      Avg.      |
> > | -------------------- | :-------: | :--------: | :---: | :---: | :------: | :---: | :-------------: |
> > | Base                 |   0.829   |   0.780   | 0.717 | 0.622 |  0.118  | 0.418 |      0.503      |
> > | ORM (Skywork)        |   0.841   |   0.787   | 0.746 | 0.661 |  0.165  | 0.441 |      0.531      |
> > | PRM (Qwen)           |   0.860   |   0.817   | 0.714 | 0.614 |  0.138  | 0.423 |      0.516      |
> > | ORM (Skywork) + MCTS |   0.854   |   0.805   | 0.746 | 0.661 |  0.193  | 0.424 |      0.538      |
> > | ORM (Trained)        |   0.854   |   0.823   | 0.746 | 0.643 |  0.214  | 0.449 |      0.549      |
> > | ORM (Trained) + MCTS |   0.860   |   0.826   | 0.743 | 0.648 |  0.221  | 0.441 |      0.550      |
> > | VM-MCTS              |   0.866   |   0.829   | 0.775 | 0.675 |  0.415  | 0.471 |      0.615      |
> > | **ReST-RL**    |   0.866   |   0.829   | 0.817 | 0.704 |  0.642  | 0.506 | **0.689** |
> >
> > **Minor issue 1: Scope of reasoning claims.**
> > We present in Table 3 our method's generalizability on math reasoning tasks (MATH, Omni-MATH). **Results demonstrate that ReST-RL and its VM have strong transferability and generalizability across unseen reasoning domains and policy checkpoints**. Note that this competitive performance enhancement on MATH/Omni-MATH is achieved without any math-specific tuning, confirming that ReST-RL extends beyond coding.
> >
> > *Table 3: Generalization of the ReST-RL approach to out-of-domain prompts (Qwen3-8B, we have supplemented these results in Section 3.4 in the main paper).*
> >
> > | Method                                 |    APPS-500    |       BCB       |      MATH      |    Omni-MATH    |
> > | -------------------------------------- | :-------------: | :-------------: | :-------------: | :-------------: |
> > | Base (0th iter.)                       |      0.118      |      0.418      |      0.780      |      0.234      |
> > | Base (0th iter.) + VM (0th iter.)      |      0.415      |      0.471      |      0.828      |      0.238      |
> > | ReST-GRPO (2nd iter.) + VM (0th iter.) |      0.630      |      0.496      |      0.862      |      0.246      |
> > | ReST-GRPO (2nd iter.) + VM (2th iter.) | **0.642** | **0.506** | **0.872** | **0.256** |
> >
> > **Minor issue 2: Figure/Table captions.**
> > We apologize for not clearly explaining the meaning of "Iter" in the table. As described in Section 3.2, "Iter" here actually refers to the number of training iterations performed on the LLM on `Q_{train}` (0th Iter. means not trained). We have updated the caption in the subsequent version to clarify this.
> >
> > Overall, we hope these clarifications demonstrate that ReST-RL is novel, robust and generalizable. Thank you again for the insightful feedback. If you feel our response has addressed some of your concerns, we sincerely hope you will adjust your rating. Thank you!

---

> ### Author Response · Authors · 2025-11-28
> **Reminder**
>
> **Dear Reviewer SYgK,**
>
> Thank you again for the time and effort you have devoted to reviewing our submission. We have provided detailed responses and additional clarifications to all reviewer comments. As the rebuttal period is approaching its final deadline, we would kindly like to remind you to check whether any further feedback or follow-up questions are needed from our side. We sincerely appreciate your consideration and would be grateful for any updates on the scores at your convenience.
>
> Thank you very much for your time.

---

### Official Review · Reviewer_8UK7 · 2025-11-01

**Soundness:** 3
**Presentation:** 3
**Contribution:** 3
**Rating:** 6
**Confidence:** 4

**Summary:**

The paper introduces ReST-RL, a two-stage reinforcement learning framework for improving code reasoning in large language models. The method unifies GRPO with a value-guided decoding strategy, addressing the low reward variance in GRPO and the annotation cost of process reward models:
- ReST-GRPO: Builds on GRPO and self-training by sampling multiple solutions per prompt, filtering data by reward variance and maximum reward, and assembling partial prompts from high-reward traces.
- VM-MCTS: Trains a Value Model (VM) using rollouts collected from MCTS with rule-based rewards. At inference, an adapted MCTS leverages VM predictions both as process signals and for solution verification (Best-of-N over VM scores).

Experiments on HumanEval, MBPP, APPS-500, and BigCodeBench using several 6-8B LLMs demonstrate that 1) ReST-GRPO outperforms naive GRPO and ReST-DPO in all benchmarks. 2) VM-MCTS surpasses PRM, ORM, and ORM-MCTS under identical 100-sample budgets 3) ReST-GRPO trains faster and sustains improvement over longer steps.

**Strengths:**

1. The framework harmonizes online RL, self-training, and MCTS-based decoding in a coherent way, showing the benefit of connecting these previously separate threads.
2. Strong empirical improvements: Both ReST-GRPO and VM-MCTS deliver measurable gains across multiple code LLMs and benchmarks. The improvements are robust under both sampling and greedy decoding.
3. Data efficiency: The entire pipeline uses ~7k prompts and no human-annotated rewards, demonstrating a favorable cost–performance trade-off.

**Weaknesses:**

1. Baseline mismatch for PRM/ORM: VM is trained on code with unit‑test targets; ORM/PRM baselines are generic (Skywork ORM; Qwen2.5 PRM).
2. Ablations on ReST‑GRPO design: No sensitivity analyses for $\sigma_0, r_0, \alpha, \beta, N$; unclear contributions of filtering vs. partial‑state restarts.
3. Reward‑shaping: GRPO training uses Eq. (8) (substring bonus, redundancy penalty) with non‑negligible weights; it’s not shown how much these influence outcomes, nor whether DAPO/GRPO baselines used identical shaping.

**Questions:**

1. How were BCB and APPS splits handled to prevent overlap between training and evaluation?
2. Could you provide results using PRM/ORM models trained on the same code dataset to ensure domain parity?
3. How sensitive is ReST-GRPO to $\sigma_0, r_0, \alpha, \beta$? Which component contributes most? variance filtering or partial-state sampling?
4. How do results change if you remove the essential‑substring and redundancy terms in Eq. (8)? Did DAPO/GRPO baselines use the exact same shaping?
5. Appendix A.8 assumes unbiased and independent estimators with equal variance. What happens empirically when the VM is biased/correlated with rewards?

---

> ### Author Response · Authors · 2025-11-21
> **Response to Reviewer 8UK7 (1/2)**
>
> Thank you for acknowledging the coherence of our two-stage framework and its empirical robustness across code LLMs. We respond to each concern below.
>
> **W1&Q2: Domain parity for baselines.**
> Your concerns regarding domain parity are profound. To eliminate domain mismatch, we trained a Qwen3-8B based ORM (denoted as ORM (Trained)) on the same coding dataset `Q_{train}` using the identical unit-test reward as the VM. The test performance comparison is shown in Table 1 below. The results show that **even when using a dedicated ORM trained on the same dataset and employing MCTS to reinforce decoding (54.9% and 55.0%), it still falls short of ReST-RL's reasoning performance (68.9%)**. This further solidifies the superiority of our approach.
>
> *Table 1: Test performance of ReST-RL and other enhanced decoding baselines (Qwen3-8B, we have supplemented it in Appendix A.7 in the main paper).*
>
> | Method               | HumanEval | HumanEval+ | MBPP | MBPP+ | APPS-500 |  BCB  |      Avg.      |
> | -------------------- | :-------: | :--------: | :---: | :---: | :------: | :---: | :-------------: |
> | Base                 |   0.829   |   0.780   | 0.717 | 0.622 |  0.118  | 0.418 |      0.503      |
> | ORM (Skywork)        |   0.841   |   0.787   | 0.746 | 0.661 |  0.165  | 0.441 |      0.531      |
> | PRM (Qwen)           |   0.860   |   0.817   | 0.714 | 0.614 |  0.138  | 0.423 |      0.516      |
> | ORM (Skywork) + MCTS |   0.854   |   0.805   | 0.746 | 0.661 |  0.193  | 0.424 |      0.538      |
> | ORM (Trained)        |   0.854   |   0.823   | 0.746 | 0.643 |  0.214  | 0.449 |      0.549      |
> | ORM (Trained) + MCTS |   0.860   |   0.826   | 0.743 | 0.648 |  0.221  | 0.441 |      0.550      |
> | VM-MCTS              |   0.866   |   0.829   | 0.775 | 0.675 |  0.415  | 0.471 |      0.615      |
> | **ReST-RL**    |   0.866   |   0.829   | 0.817 | 0.704 |  0.642  | 0.506 | **0.689** |
>
> **W2&Q3: Ablations on ReST-GRPO design.**
> In Table 2, we present an ablation for ReST-GRPO showing that our default setting (as illustrated in Appendix A.5 and A.6, here represented by `r₀=0.9`) for sampling delivers the best variety of rewards, validating the **filtering and partial sampling** methodology. From the perspective of reward variance (learning signal), we can see the significant importance of filtering operations (0.104 → 0.148, related to `σ₀` and `r₀`), but subsequent high-value trajectory selection and partial sampling are crucial as well (0.148 → 0.168, related to `r₀`,  `α` and `β`). Aside from reward variance, our sampling strategy utilizes the high reward traces to automatically form valuable partial samples, we believe this enables the LLM to **thoroughly explore important action subspaces** during optimization, which leads to observed further enhancements.
>
> *Table 2: Ablation on hyperparameters for the ReST-GRPO algorithm from the perspective of reward standard deviation (Qwen3-8B, we have supplemented it in Appendix A.7 in the main paper).*
>
> | Sampling Method | σ₀=0, β=0 | α → 0(0.01) | r₀=0.1 | r₀=0.5 | r₀=0.9         |
> | :-------------: | :----------: | :-----------: | :-----: | :-----: | --------------- |
> |     Mean SD     |    0.104    |     0.148     |  0.157  |  0.163  | **0.168** |
> |    Median SD    |    0.000    |     0.120     |  0.111  |  0.124  | **0.130** |

---

> > ### Author Response · Authors · 2025-11-21
> > **Response to Reviewer 8UK7 (2/2)**
> >
> > **W3&Q4: Reward shaping consistency.**
> > We apologize for not explicitly stating the scope of application for this reward shaping. In fact, all compared methods (GRPO, DAPO) already use the identical shaping terms. We introduced reward shaping because certain coding tasks require specific formatting for output code blocks, and some models (such as Llama-3) exhibit limited capability in adhering to these formatting rules. We observed that training solely based on the core reward leads to slow improvements due to formatting issues for these models. Adding a small output shaping reward (experiments used 1e-3 and 1e-6 as coefficients) mitigates this issue. However, it's important to note that this reward shaping merely encourages LLM output specification and does not substantially alter their final reasoning performance or accuracy.
> >
> > **Q1: Dataset splits.**
> > We ensure that the training and test sets reside in distinct partitions, with no identical prompts appearing in both the training and test portions. Throughout the entire ReST-RL training phase, test data remains invisible to both the LLM and the VM.
> >
> > **Q5: Assumption on unbiased VM.**
> > Appendix A.8 presents the desirable properties of VM estimates under ideal conditions. However, we must acknowledge that actual VMs are not perfectly unbiased, stemming from the variance and bias inherent in the rollout itself, as well as deficiencies within the model network. This bias may introduce noise, potentially affecting the accuracy of VM-MCTS estimates to some extent. Nevertheless, convergence guarantees that as the number of rollout samples approaches infinity, the trained VM network achieves the ideal unbiased estimate. Therefore, even with existing biases in practice, Appendix A.8 still provides meaningful insights.
> >
> > In conclusion, we hope these clarifications demonstrate that ReST-RL is both robust and extensible. Thank you again for the insightful feedback. If you feel our response has addressed some of your concerns, we sincerely hope you will adjust your rating. Thank you!

---

> ### Author Response · Authors · 2025-11-28
> **Reminder**
>
> **Dear Reviewer 8UK7,**
>
> Thank you again for the time and effort you have devoted to reviewing our submission. We have provided detailed responses and additional clarifications to all reviewer comments. As the rebuttal period is approaching its final deadline, we would kindly like to remind you to check whether any further feedback or follow-up questions are needed from our side. We sincerely appreciate your consideration and would be grateful for any updates on the scores at your convenience.
>
> Thank you very much for your time.

---

### Official Review · Reviewer_PFgs · 2025-11-03

**Soundness:** 3
**Presentation:** 3
**Contribution:** 2
**Rating:** 6
**Confidence:** 4

**Summary:**

The paper proposes ReST-RL, a two-stage reinforcement learning (RL) framework to improve code reasoning in large language models (LLMs). Stage 1, ReST-GRPO, augments Group Relative Policy Optimization (GRPO) with an improved ReST-style self-training loop that filters and assembles high-reward partial solutions to increase reward variance and training efficiency. Stage 2, VM-MCTS, trains a Value Model (VM) via Monte-Carlo Tree Search (MCTS) without extra annotations, then uses the VM to guide MCTS decoding at test time. Extensive experiments on HumanEval, MBPP, APPS-500 and BigCodeBench show that ReST-RL outperforms baselines, while using limited training data.

**Strengths:**

- Introduces a principled combination of self-training (ReST sampling/assembly) with GRPO and a value-model guided MCTS at test time. The combination and the particular sampling/filtering heuristics (σ0, r0, exponential sampling with α) appear novel in this exact form.
- Solid empirical coverage: 6 benchmarks, 4 base code-LLMs, plus general LLMs (Llama-3, Qwen3) with both pass@1 and test-case-average metrics.
- Mathematical justification (Appendix A.8) proves that value-based MC rollouts produce lower-variance value estimates than full-trace rollouts and supports design choice.
- Algorithms are presented via clear pseudocode boxes (Alg. 1–3).
- Demonstrates that nontrivial improvements are achievable with few training prompts, suggesting data-efficient RL for code is possible.
- Provides a plug-in decoding module (VM-MCTS) that can be applied on top of any base policy without re-training.

**Weaknesses:**

- Missing statistical significance tests. Reported gains are single-run (no std-dev). With pass@1 often changing by ≤3 points between iterations (Table 1), observed differences may be within random noise. For RL-style training this is critical.
- The paper gives some VM-MCTS hyperparameters (T, n, c) for experiments but lacks a centralized hyperparameter table and justification of choices.
- Value model generalization unclear: VM is trained on the same coding prompts; no out-of-domain (OOD) evaluation (e.g., math reasoning) or scaling curve with more prompts.

**Questions:**

- Can you supply an ablation showing ReST-GRPO without the partial-state assembly (β=0 or α→0) to demonstrate the added value of partial state sampling?
- Does the value model retain accuracy when deployed on coding domains not seen during training?
- What is the computational overhead of VM-MCTS at test time (rollouts, latency, memory) compared to Best-of-N baselines?
- How does ReST-RL perform with non-code reasoning tasks (e.g., GSM8K) to support broader applicability?
- performance vs. GPU-hours for ReST-GRPO vs naive GRPO / DAPO to validate efficiency claims.(Fig.3 a claims but lacks compute axis)

---

> ### Author Response · Authors · 2025-11-21
> **Response to Reviewer PFgs (1/2)**
>
> We sincerely appreciate your thorough evaluation—especially your recognition of the principled combination of self-training with GRPO and the extensive empirical coverage. Below we address each concern in turn.
>
> **W1: Missing statistical significance tests.**
> Your concerns about the significance of the results are reasonable. To estimate the mean and std-dev accurately, we conducted four independent replicate experiments under identical conditions mentioned in Section 3.2. We now report self-train results with std-dev for Qwen3-8B on every benchmark. As shown in Table 1, ReST-GRPO maintains clear margins (>5% Avg. , clearly not within random noise) over all other baselines and across iterations, indicating its significant improvement.
>
> *Table 1: Self-training results of Qwen3-8B with statistics (we have supplemented these results in Appendix A.7 in the main paper).*
>
> | Training Method                 |  HumanEval  |  HumanEval+  |     MBPP     |    MBPP+    |   APPS-500   |     BCB     |          Avg.          |
> | ------------------------------- | :----------: | :----------: | :----------: | :----------: | :----------: | :----------: | :--------------------: |
> | Base (0th iter.)                | 0.829±0.018 | 0.780±0.019 | 0.717±0.003 | 0.622±0.002 | 0.118±0.003 | 0.418±0.007 |      0.503±0.008      |
> | ReST-DPO (1st iter.)            | 0.854±0.013 | 0.799±0.006 | 0.730±0.005 | 0.627±0.011 | 0.152±0.015 | 0.434±0.010 |      0.523±0.011      |
> | GRPO (1st iter.)                | 0.799±0.018 | 0.750±0.030 | 0.754±0.016 | 0.651±0.013 | 0.346±0.006 | 0.439±0.004 |      0.566±0.012      |
> | **ReST-GRPO (1st iter.)** | 0.872±0.012 | 0.817±0.006 | 0.780±0.005 | 0.672±0.003 | 0.377±0.008 | 0.469±0.013 |      0.604±0.009      |
> | ReST-DPO (2nd iter.)            | 0.841±0.006 | 0.811±0.012 | 0.741±0.003 | 0.653±0.007 | 0.153±0.011 | 0.429±0.007 |      0.526±0.008      |
> | GRPO (2nd iter.)                | 0.829±0.006 | 0.787±0.006 | 0.757±0.006 | 0.667±0.014 | 0.403±0.007 | 0.436±0.004 |      0.590±0.007      |
> | **ReST-GRPO (2nd iter.)** | 0.860±0.012 | 0.805±0.012 | 0.802±0.006 | 0.690±0.010 | 0.565±0.022 | 0.476±0.007 | **0.655**±0.012 |
>
> **W2&Q1: Lack of centralized hyperparameter table / Justification and ablation of hyperparameters.**
> In Appendix A.5 and A.6, we already specified the hyperparameters employed. Here in Table 2 and Table 3, we seperately list the main hyperparameters for ReST-GRPO and VM-MCTS to avoid confusion. In Table 4 we present an ablation for ReST-GRPO showing that our default setting (denoted by `r₀=0.9`, see configurations of all other hyperparameters in Table 3) for sampling delivers the best variety of rewards, validating the **filtering and partial sampling** methodology. Our sampling strategy utilizes the high reward traces to automatically form valuable partial samples, enabling the LLM's thorough exploration of action subspaces during optimization. As for the hyperparameters of VM-MCTS, we empirically select their values based on simple observations of the test results and considerations of resource constraints. Under our setting, the assisted LLMs already exhibit significant enhancements in reasoning accuracy, though according to Figure 3(b) increasing `T` or `n` is likely to further improve the final score.
>
> *Table 2: Main hyperparameters for the VM-MCTS method (we have supplemented it in Appendix A.9 in the main paper).*
>
> | Hyperparameter |         T         |              n              |          c          |                   ϵ                   |
> | -------------- | :---------------: | :-------------------------: | :------------------: | :------------------------------------: |
> | Meaning        | Search iterations | Number of branches expanded | Exploration constant | Constant avoiding zero-division in UCT |
> | Default Value  |        30        |              5              |         0.4         |                  0.1                  |
>
> *Table 3: Main hyperparameters for the ReST-GRPO method (we have supplemented it in Appendix A.9 in the main paper).*
>
> | Hyperparameter |             N             |     σ₀     |       r₀       |            α            |      β      |
> | -------------- | :------------------------: | :----------: | :--------------: | :----------------------: | :----------: |
> | Meaning        | Number of solution samples | SD threshold | Reward threshold | Sampling exponent factor | Sample ratio |
> | Default Value  |             30             |     0.05     |       0.9       |           0.95           |     0.5     |

---

> ### Author Response · Authors · 2025-11-21
> **Response to Reviewer PFgs (2/2)**
>
> *Table 4: Ablation on hyperparameters for the ReST-GRPO algorithm from the perspective of reward standard deviation (Qwen3-8B, we have supplemented it in Appendix A.7 in the main paper).*
>
> | Sampling Method | σ₀=0, β=0 | α → 0(0.01) | r₀=0.1 | r₀=0.5 | r₀=0.9         |
> | :-------------: | :----------: | :-----------: | :-----: | :-----: | --------------- |
> |     Mean SD     |    0.104    |     0.148     |  0.157  |  0.163  | **0.168** |
> |    Median SD    |    0.000    |     0.120     |  0.111  |  0.124  | **0.130** |
>
> **W3&Q2&Q4: Method generalization beyond the training domain.**
> It is indeed crucial to confirm the generalization capability of the ReST-RL method on out-of-domain data. In our original paper, we included Humaneval(+) and MBPP(+) as test benchmarks, whose relevant data never appeared during training. The experimental results demonstrate that the ReST-RL method performs well on these test datasets (Table 2 and Figure 4 in the paper). Furthermore, Table 5 examines its generalizability on math reasoning tasks (MATH, Omni-MATH). Results demonstrate that **ReST-RL and its VM have strong transferability and generalizability across unseen reasoning domains and policy checkpoints**. Note that this competitive performance enhancement on MATH/Omni-MATH is achieved without any math-specific tuning, confirming that ReST-RL extends beyond coding.
>
> *Table 5: Generalization of the ReST-RL approach to out-of-domain prompts (Qwen3-8B, we have supplemented these results in Section 3.4 in the main paper).*
>
> | Method                                 |    APPS-500    |       BCB       |      MATH      |    Omni-MATH    |
> | -------------------------------------- | :-------------: | :-------------: | :-------------: | :-------------: |
> | Base (0th iter.)                       |      0.118      |      0.418      |      0.780      |      0.234      |
> | Base (0th iter.) + VM (0th iter.)      |      0.415      |      0.471      |      0.828      |      0.238      |
> | ReST-GRPO (2nd iter.) + VM (0th iter.) |      0.630      |      0.496      |      0.862      |      0.246      |
> | ReST-GRPO (2nd iter.) + VM (2th iter.) | **0.642** | **0.506** | **0.872** | **0.256** |
>
> **Q3: VM-MCTS computational overhead.**
>
> In Table 5 we display the decoding cost of VM-MCTS compared to conventional Best-of-N, where `N` is the total number of sampled traces. Combined with Figure 3(b), we can observe that **VM-MCTS significantly outperforms Best-of-N across all sampling sizes while requiring fewer sampled tokens**, owing to its ability to perform rollouts from partial samples.
>
> *Table 5: Average decoding computation cost (k tokens per prompt, CodeQwen, we have supplemented these results in Figure 3 in the main paper).*
>
> | Avg. Tokens (k) / prompt | N=5 | N=10 | N=20 | N=50 | N=100 |
> | ------------------------ | :--: | :--: | :--: | :---: | :---: |
> |  Best-of-N                | 1.29 | 2.57 | 5.14 | 12.80 | 25.56 |
> | VM-MCTS               | 1.28 | 2.54 | 4.96 | 12.06 | 22.43 |
>
> **Q5: Training efficiency of ReST-GRPO.**
>
> Table 6 shows the GPU-hours required for every 1k training samples for ReST-GRPO and naive GRPO. Due to its pre-train filtering and partial sampling design, ReST-GRPO attains an **approximately 20% reduction on training time, while achieving better training results** (Figure 3(a) in our paper). This confirms our claim regarding efficiency.
>
> *Table 6: Training GPU-hour accounting (Llama-3-8B, we have supplemented these results in Figure 3 in the main paper).*
>
> | GPU Hours / k samples | 1   |  2  |  3  |  4  |  5  |  6  |  7  |  8  |  9  |  10  |
> | --------------------- | --- | :-: | :-: | :-: | :--: | :--: | :--: | :--: | :--: | :--: |
> | GRPO                  | 209 | 415 | 612 | 819 | 1035 | 1251 | 1454 | 1654 | 1865 | 2081 |
> | ReST-GRPO             | 163 | 325 | 482 | 640 | 818 | 990 | 1168 | 1344 | 1515 | 1672 |
>
> Overall, we hope these clarifications demonstrate that ReST-RL is both practical and generalizable. Thank you again for the insightful feedback. If you feel our response has addressed some of your concerns, we sincerely hope you will adjust your rating. Thank you!

---

> ### Author Response · Authors · 2025-11-28
> **Reminder**
>
> **Dear Reviewer PFgs,**
>
> Thank you again for the time and effort you have devoted to reviewing our submission. We have provided detailed responses and additional clarifications to all reviewer comments. As the rebuttal period is approaching its final deadline, we would kindly like to remind you to check whether any further feedback or follow-up questions are needed from our side. We sincerely appreciate your consideration and would be grateful for any updates on the scores at your convenience.
>
> Thank you very much for your time.

---

### Author Response · Authors · 2025-11-26
**Overview of Review and Rebuttal**

We appreciate all reviewers' careful assessments: reviewer PFgs highlighted the principled fusion of self-training with GRPO and the broad empirical validation; reviewer 8UK7 recognized the coherence and robustness of the two-stage framework across several base models; reviewer SYgK emphasized the intuitive appeal and empirical strength of the partial-state completion strategy; reviewer zamV praised the originality of unifying offline self-training, online RL, and value-guided tree search along with the clarity of our presentation.

To address the major concerns (grouping related points), we have made the following additions or modifications:

- **Statistical significance:** added replicate self-training runs on every benchmark, reporting means/standard deviations to demonstrate the stability of ReST-GRPO gains.
- **Hyperparameters and ablations:** consolidated the core hyperparameters for ReST-GRPO and VM-MCTS and provided reward variance ablations over sampling/threshold settings, showing how high-reward filtering and partial sampling drive performance.
- **Domain parity and verifier fairness:** re-trained an ORM on the exact VM dataset and compared all verifiers under identical Best-of-N/MCTS settings, confirming that ReST-RL still delivers the largest lift.
- **Generalization and VM transfer:** included cross-domain (Humaneval(+), MBPP(+), MATH, Omni-MATH) and cross-iteration evaluations to show the VM remains effective as the policy evolves and when applied to out-of-domain reasoning.
- **Training/inference efficiency:** reported GPU-hour accounting plus VM-MCTS vs. Best-of-N token usage to justify our efficiency claims.
- **VM bias discussion:** analyzed mixed-iteration deployments to show the VM mitigates rollout variance even when mildly biased and clarified the theoretical assumptions in the appendix.
- **Data splits and reward shaping:** documented the strict dataset partitioning and the shared output-shaping terms across all methods to avoid ambiguity.
- **Rule-based reward scope:** explicitly discussed why test-case rewards suffice for our code focus while noting ongoing exploration of lightweight learned rewards.

Overall, we believe we have resolved the main issues. We hope these updates allow reviewers and ACs to assess ReST-RL more precisely, and we thank you again for the thoughtful feedback. We hope that our resolution of the issues will be reflected in the scores and the final assessment. Should any unresolved issues remain, we sincerely request that reviewers promptly point them out. Thank you!

---

### Meta-Review · Area_Chair_mLuw · 2026-01-06

**Summary:**

The initial reviews for ReST-RL were polarized, ranging from "Marginally Above Threshold" (PFgs, 8UK7) to "Reject" (SYgK, zamV). The primary technical and procedural concerns were: 1. Statistical Rigor: A lack of significance tests and standard deviations in initial results made it difficult to distinguish real gains from random noise. 2. Computational Efficiency: Reviewers questioned the "data-efficient" and "training-efficient" claims. 3. Novelty vs. Existing Paradigms: Reviewers questioned the novelty of combining ReST with GRPO and how it differs from contemporary R1-style reinforcement learning or MCTS baselines. 4. Generalizability and Out-of-Distribution (OOD) Performance: Concerns were raised about whether the Value Model (VM) and policy would generalize beyond coding to other reasoning tasks (e.g., Math).

**Reviewer Concerns:**

Addressed Concerns:
1. Statistical Significance: The authors provided four independent replicate experiments with mean and standard deviation.
2. Generalization to Math: Authors added results for MATH and Omni-MATH benchmarks. Without math-specific tuning, ReST-RL improved accuracy from 78.0% to 87.2%, proving the framework's broad applicability to general reasoning beyond code.
3. Efficiency Breakdown: Detailed GPU-hour accounting was provided, showing that ReST-GRPO achieves a ~20% reduction in training time compared to naive GRPO while yielding better results.

Outstanding Concerns:
1. Complexity of Two-Stage Paradigm: While robust, the interaction between ReST-GRPO (training) and VM-MCTS (decoding) remains complex. Some reviewers still find the decoupling of policy and value training to be a hurdle for simple deployment compared to monolithic R1-style GRPO.
2. Hyperparameter Sensitivity: Although the authors provided a centralized table and some ablations for $\sigma_0$ and $r_0$, the sensitivity of the $T$ (iterations) and $n$ (branches) in MCTS decoding remains somewhat empirically driven.

**Reviewer Scores:**

PFgs	6 maintain score.
8UK7	6 maintain score.
SYgK	2->4 Authors addressed the distribution shift and novelty concerns.
zamV	2->4 Detailed cost breakdown and R1/PSPO comparison addressed rejection criteria.

---

### Decision · Program_Chairs · 2026-01-26

Reject